# Simultaneous enhancement of multiple functional properties using evolution-informed protein design

Benjamin Fram[1,2] ✉, Yang Su[1,14], Ian Truebridge[3,4,10,14], Adam J. Riesselman[1,5], John B. Ingraham[1], Alessandro Passera[2,11], Eve Napier[6], Nicole N. Thadani[1,12], Samuel Lim[1], Kristen Roberts[7], Gurleen Kaur[7], Michael A. Stiffler[2,13], Debora S. Marks ⓘ[1,8], Christopher D. Bahl ⓘ[3,4,10], Amir R. Khan[6,9], Chris Sander[1,2,8] & Nicholas P. Gauthier ⓘ[1,2,8] ✉

A major challenge in protein design is to augment existing functional proteins with multiple property enhancements. Altering several properties likely necessitates numerous primary sequence changes, and novel methods are needed to accurately predict combinations of mutations that maintain or enhance function. Models of sequence co-variation (e.g., EVcouplings), which leverage extensive information about various protein properties and activities from homologous protein sequences, have proven effective for many applications including structure determination and mutation effect prediction. We apply EVcouplings to computationally design variants of the model protein TEM-1 $\beta$-lactamase. Nearly all the 14 experimentally characterized designs were functional, including one with 84 mutations from the nearest natural homolog. The designs also had large increases in thermostability, increased activity on multiple substrates, and nearly identical structure to the wild type enzyme. This study highlights the efficacy of evolutionary models in guiding large sequence alterations to generate functional diversity for protein design applications.

As proteins become increasingly useful across a range of fields including medicine and industry, there is a growing need for designed proteins with optimized characteristics, such as elevated thermostability, higher binding affinity, or increased catalytic activity. Natural proteins are often used as starting points for the development of useful proteins, which can then be engineered as high-performance, task-specific tools. However, efficiently mutating enzymes to yield optimized variants is exceedingly difficult, and randomly mutating enzymes almost always leads to loss of performance, which decreases considerably with every additional mutation[1]. Information-based 'rational' engineering can avoid performance loss, but is generally limited to a very small number of sequence changes. One approach to protein engineering, directed evolution, makes use of iterative rounds of mutagenesis followed by selection to optimize a specific property

[1]Department of Systems Biology, Harvard Medical School, Boston, MA, USA. [2]Department of Data Sciences, Dana-Farber Cancer Institute, Boston, MA, USA. [3]Institute for Protein Innovation, Boston, MA, USA. [4]Division of Hematology/Oncology, Boston Children's Hospital, Harvard Medical School, Boston, MA, USA. [5]Program in Biomedical Informatics, Harvard Medical School, Boston, MA, USA. [6]School of Biochemistry and Immunology, Trinity College Dublin, Dublin 2, Ireland. [7]Selux Diagnostics Inc., 56 Roland Street, Charlestown, MA, USA. [8]Broad Institute of MIT and Harvard, Cambridge, MA, USA. [9]Division of Newborn Medicine, Boston Children's Hospital, Boston, MA, USA. [10]Present address: AI Proteins, Boston, MA, USA. [11]Present address: Research Institute of Molecular Pathology (IMP), Vienna BioCenter (VBC), Campus-Vienna-Biocenter 1, 1030 Vienna, Austria. [12]Present address: Apriori Bio, Cambridge, MA, USA. [13]Present address: Dyno Therapeutics, 343 Arsenal Street, Watertown, MA, USA. [14]These authors contributed equally: Yang Su, Ian Truebridge. ✉e-mail: benjamin.fram.research@gmail.com; nicholas.gauthier.research@gmail.com

like activity or thermostability. However, increased random mutation count overwhelmingly negatively impacts fitness[1], limiting the number of amino acid changes that can be introduced while still maintaining a reasonable number of functional variants. This stepwise incremental selection strategy is often effective at finding sequences with improved properties with a limited number of mutations. The introduction of many simultaneous changes to a protein's primary sequence is likely required to diversify and optimize multiple desirable properties, and new methods that enable such large changes in primary sequence are needed. Computational design strategies[2-7], which account for the complexity of how each mutated residue interacts with all other residues, are likely required to maintain function when introducing more than a handful of mutations.

An evolution-informed computational protein design strategy may provide a means to generate many changes in primary sequence, enabling the exploration of diverse structural and functional properties. Evolutionary models that account for complex selective conditions over millions of years by learning meaningful constraints on function from related sets of homologous protein sequences[5,8-10] have been shown to recapitulate core aspects of protein biology, such as 3D structure[8,11,12], protein stability[10,13,14], conformational state[15-17] and the effects of mutations on protein fitness[2,10,13,18-20]. Some of these models have been used for protein design, generating mutated proteins from a wild type scaffold that maintain structure and/or function[2,5-7,21-23]. Evolutionary couplings (EVcouplings) models are a specific instance of evolutionary models based on residue site- and pairwise dependencies in natural sequence variation[8,10,24]. These models are unsupervised, inferring sequence constraints characteristic of a functional space and quantifying fitness differences between variants without experimentally measured phenotype labels. To make use of these discriminative models for protein design, a sampling algorithm is used to iteratively

generate variant sequences that are chosen to optimize a fitness function.

The TEM-1 $\beta$-lactamase model system has been extensively used to study protein evolution[7,9,25-28]. $\beta$-lactamases are a class of enzymes that are produced by bacteria in order to provide resistance to bactericidal $\beta$-lactam antibiotics through hydrolysis of their core $\beta$-lactam ring. Many bench biologists are familiar with the use of TEM-1 $\beta$-lactamase as a marker for successful transformation, in which selection of functional TEM-1-containing plasmids is as simple as growth in a $\beta$-lactam antibiotic like ampicillin. Due to experimental tractability, many publications report the effects of mutations on TEM-1 function and stability, including several studies using deep mutational scans[25,27,29]. Other studies have described the exponential decrease in TEM-1 function when subjected to multiple mutations, with the cumulative effect of 10 random mutations completely abrogating enzyme activity[1].

In this work we investigate whether evolutionary models of sequence co-variation can be used to design enzyme variants that contain many changes to the target sequence while maintaining function. In addition, we test whether making large jumps in the primary amino acid sequence can lead to augmented protein properties such as increased thermostability, increased activity and broadening of available substrates, and investigate the implications of these mutations on protein 3D structure.

## Results
### Protein design and testing workflow
Protein design from discriminative models such as EVcouplings[10,30] is a multistep process (Fig. 1). A multiple sequence alignment (MSA) of homologous proteins is generated for the protein of interest, which is then used to generate a site and pairwise evolutionary model. This maximum entropy model quantifies evolutionary constraints and is

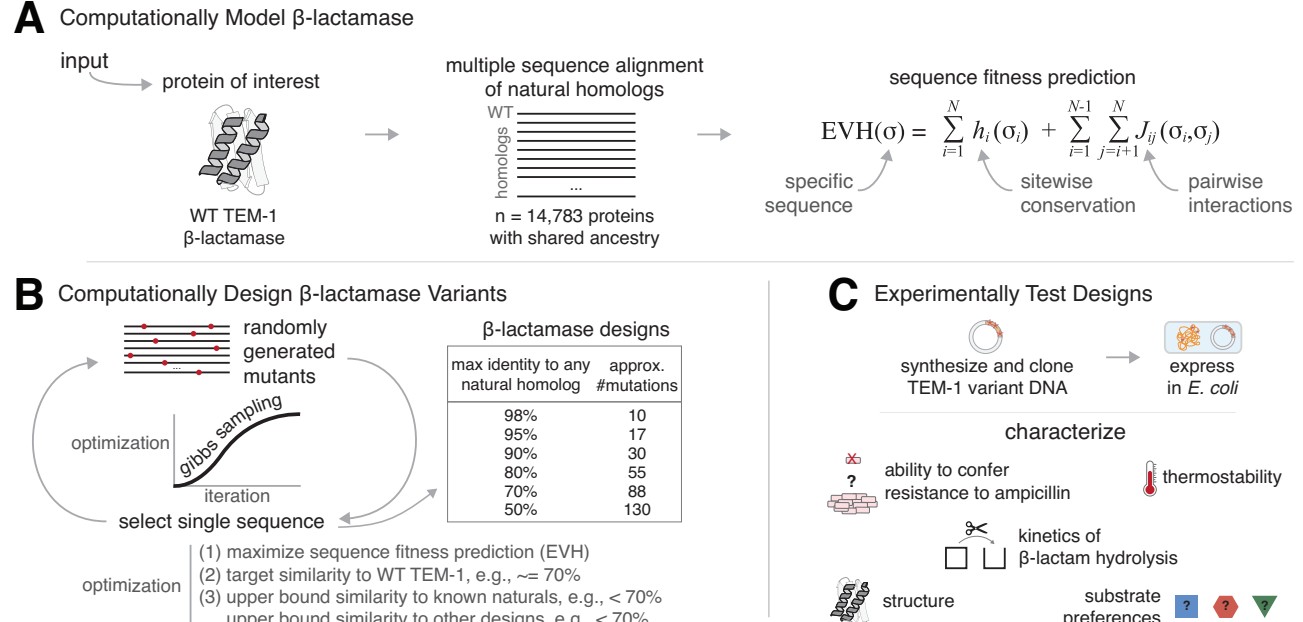

**Fig. 1 | The $\beta$-lactamase variant design process.** Strategy applied to generate and test design variants using an evolution-informed statistical model of the $\beta$-lactamase protein family. [**A**, Computationally Model $\beta$-lactamase] WT TEM-1 $\beta$-lactamase was used to generate a multiple sequence alignment that was used as input to derive an EVcouplings maximum entropy model. The predicted fitness (EVH) for any sequence ($\sigma$) can be calculated as the sum of coupling terms ($J_{ij}$) between every pair of residues as well as site-wise conservation terms ($h_i$). [**B**, Computationally

Design $\beta$-lactamase Variants] Design variants are generated by using Gibbs sampling to iteratively optimize an objective function that takes into account EVH and sequence similarity to WT TEM-1, to natural homologs, as well as to the other designed sequences. [**C**, Experimentally Test Designs] Designs were synthesized, cloned into plasmids, expressed in *E. coli*, and several experimental protocols were performed to characterize each design including cell-based activity assays, biochemical kinetics assessment, and structure determination and analysis.

parameterized by both site-specific ($h_i$) and pairwise or epistatic ($J_{ij}$) constraints (where i and j indicate amino acid positions) with minimal spurious information in the parameter set. The predicted fitness of a specific sequence ($\sigma$) is defined as the statistical energy (evolutionary Hamiltonian or EVH, Fig. 1A). To have confidence that the model is capable of generating functional sequences, model quality is assessed here by comparing predictions to known biological properties, e.g., recapitulating known structural contacts common to the protein family and/or the known effects of point mutations on protein fitness of individual sequences.

Designed sequences are generated using a sampling algorithm, (e.g., Markov Chain Monte Carlo or Gibbs sampling, Fig. 1B) that optimizes EVH fitness of each entire sequence and satisfies user-specified sequence distance constraints. Conceptually, a single design variant is generated from an iterative process in which a random starting sequence is mutated over and over with the identity of the retained mutations chosen to optimize a function that includes parameters such as predicted fitness and sequence distance to target or homologous proteins. Testing of designs is dependent on the protein of interest, and can include cellular biological activity assays or detailed biochemical characterization after protein purification (Fig. 1C). For this work we used the TEM-1 $\beta$-lactamase model system, which is highly tractable for high-throughput experimental analysis in bacterial culture, and has been well studied for many purposes including evolvability[9,31,32], design[7], and the fitness effect of point mutations[9,27,28].

## Diverse design variants generated from model of protein evolution

We generated an EVcouplings model[8,10] of the $\beta$-lactamase protein family using a multiple sequence alignment of 14,793 sequences compiled by the jackhmmer[33] sequence search and alignment tool, seeded with wild type TEM-1 (WT TEM-1; UniProt P62593; bitscore cutoff = 0.5*length, Neff = 3757). Alignment depth was selected to be largely composed of bona fide $\beta$-lactamases. Analysis of the model revealed that over 80% of the top L predicted residue-residue interactions ("evolutionary couplings", where L is the length of aligned WT TEM-1 residues) match structural contacts in a known 3D structure of WT TEM-1 solved using X-ray crystallography (PDB: 1XPB[34], Fig. 2A). In addition, mutation effect prediction from the model (EVH), for single residue variants, is positively correlated with a published deep mutational scan of WT TEM-1 (replicate 1: $n = 4788$ with Spearman = 0.717; replicate 2: $n = 4769$ with Spearman = 0.702; Supplementary Fig. S1)[25]. These data indicate that the model is able to capture both the structure and functional sequence dependence of TEM-1 $\beta$-lactamase.

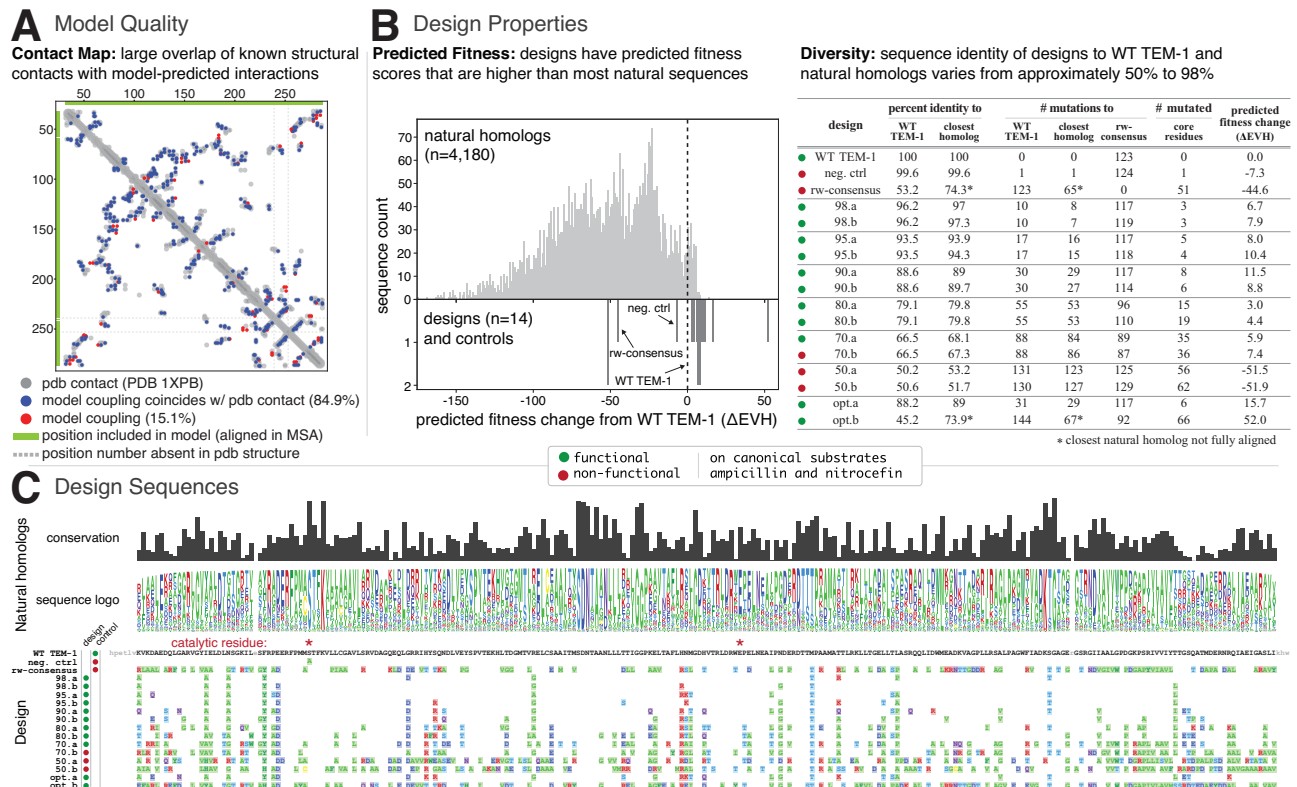

**Fig. 2 | Quality of computational model and properties of experimentally tested design variants.** [**A**, Model Quality] The computational model was evaluated for quality by comparison to published metrics of $\beta$-lactamase structure and function. Predicted residue-residue interactions (top L model couplings that are at least 5 positions apart in primary sequence) are compared with known structural contacts of WT TEM-1 as determined by X-ray crystallography (PDB: 1XPB[34]). Dots indicate contacts either in the model (couplings) or PDB structure. Green bar on y-axis (left) and x-axis (top) indicates positions that are aligned in the MSA used for model inference, and are therefore included in the model ($n = 252$). Dashed gray bars indicate position numbers that are skipped in the PDB structure (positions 239 and 253). Over 80% of the top predicted interactions are structural contacts. [**B**, Design Properties] (Predicted fitness, left): Predicted fitness changes ($\Delta$EVH) between WT TEM-1 and designs (bottom) as well as the natural homologs of $\beta$-

lactamases (top, from the multiple sequence alignment used to generate the model). (Diversity, right): Table of similarities and general properties of each design. Identity to natural homologs was determined using blastp (Methods). Core residues were defined as having a relative surface accessibility (ACC field of DSSP analysis of PDB 1XPB divided by residue size) of less than 0.2. [**C**, Design Sequences] Multiple sequence alignment (MSA) of all designs, with amino acid changes relative to WT TEM-1 colored by new residue properties (standard colors: green, hydrophobic and glycine (G); blue, negative charge; red, positive charge; light blue, polar). Conservation and logo for each position in the multiple sequence alignment of natural homologs are displayed above the design sequences. Empty columns in the logo and lowercase letters in the WT TEM-1 sequence indicate unaligned positions in the MSA and model. Source data are provided in the Source Data file.

WT TEM-1 design variants were generated algorithmically. To begin, a random amino acid sequence of length L was generated separately for each design. Batch Gibbs sampling (simultaneous optimization to all designs at each cutoff) was applied to each sequence: at each iteration, a random position was chosen for mutation and the identity of the amino acid selected to persist to the next round was determined by sampling from the conditional probability distribution of residues at that position proportional to a three-term objective function. The objective function aims to (a) maximize predicted fitness (EVH) while (b) constraining to a target sequence identity relative to WT TEM-1 and (c) enforcing an upper bound on the sequence identity relative to known natural homologs and other design variants. For example, a target of 70% results in a design that has ~70% sequence identity with WT TEM-1 and at most a 70% sequence identity with any natural homolog or other design. We generated sequences with varying sequence identity targets (98%, 95%, 90%, 80%, 70%, and 50%). Of the six sequences that were generated at each threshold, two were randomly selected for experimental testing. The sequences assayed are referred to hereafter as "98.a, 98.b, 95.a...50.a, 50.b," corresponding to their respective sequence identity threshold and an arbitrary secondary identifier. In addition, two sequences were optimized solely towards generating the highest predicted fitness (unconstrained by sequence identity): (1) The opt.a sequence was generated in a greedy manner that, starting with WT TEM-1, iteratively added mutations with the top predicted fitness until there were no additional predicted positive mutations. (2) The opt.b sequence was selected using parallel tempering[35,36] that, as applied, is a global sampling method that attempts to find the sequence with the maximum predicted fitness (Methods).

We next examined general properties of the designs (Fig. 2). Controls include WT TEM-1, a consensus sequence that contains the most represented amino acid at each position in the frequency-reweighted alignment used for model generation (rw-consensus), and a catalytically-inactive negative control where the catalytic residue Ser70 (Ser68 in UniProt numbering) was mutated to alanine (S70A, neg. ctrl)[25,37]. The designs all had a predicted fitness (EVH) higher than WT TEM-1 except for 50.a and 50.b (Fig. 2B). All of the designs had much higher predicted fitness than randomly introducing mutations into the WT TEM-1 sequence (Supplementary Fig. S2). As expected due to distance constraints imposed by the sequence generation objective function, most of the designs were more similar to WT TEM-1 than to sequences in the MSA used for model inference (Supplementary Fig. S3). While a given position was frequently altered across multiple designs, the identity of the amino acid change often varied (Fig. 2C). In general and as expected, the algorithm tended to change positions that were variable rather than conserved in the multiple sequence alignment (Fig. 2C, Supplementary Fig. S4A, and Supplementary Fig. S5). Although the algorithm did make mutations to residues in the core of the protein (Table in Fig. 2B), the algorithm tended to mutate positions that were more surface accessible than non-mutated positions (Supplementary Fig. S4B, Supplementary Fig. S5, and Supplementary Fig. S6). In addition, positions mutated in the designs generally had fewer overall interactions with other positions in WT TEM-1 (PDB: 1XPB) compared with positions that were not mutated (Supplementary Fig. S4C). An overview of each design, including the number of amino acid changes relative to WT TEM-1 and closest natural homolog as well as some general properties of mutated positions (e.g., number of mutations at core residues) can be found in Fig. 2B.

### The majority of designs confer resistance to ampicillin in bacteria and are able to hydrolyze ampicillin and nitrocefin in biochemical assays

To assess the effect of the many mutations in each design on function, the sequences were synthesized and inserted after the WT TEM-1 promoter and N-terminal signal peptide, transformed into *E. coli*, and assayed for growth in the presence of ampicillin. Using a Clinical and Laboratory Standards Institute (CLSI) broth microdilution assay, we quantified the minimum inhibitory concentration (MIC) of ampicillin required to completely abrogate the growth of bacteria that express the designed variant (Methods). The results are shown in Fig. 3A. Of the 14 designed variants, 11 conferred resistance to ampicillin. Eight had equal or increased MICs compared with WT TEM-1, three had a decreased MIC, and three had the same MIC as the catalytically-inactive negative control (neg. ctrl). In at least one replicate, several of the designs (98.b, 95.a, 95.b, 90.b, and 80.a) grew on the maximum tested concentration of ampicillin (4096 μg/mL), which completely inhibited growth of WT TEM-1. Bacteria expressing the rw-consensus sequence were unable to grow in any concentration of ampicillin above the MIC of the negative control. The two designs that were optimized solely for predicted fitness (i.e., distance unconstrained) both conferred resistance to ampicillin, with the greedy optimized sequence (opt.a) having a MIC equal to WT TEM-1, and the parallel tempering optimized sequence (opt.b) had a large decrease in MIC (roughly 100X less). In summary, nearly all of the designs were able to confer resistance to ampicillin including a design with 84 mutations relative to its closest natural homolog (70.a) and two designs with over 50 mutations (80.a and 80.b).

In addition to assessing MIC with the broth microdilution assay, we applied and obtained similar results using two additional independent antibiotic-resistance assays to determine the designs' resistance to ampicillin in cells: MIC determination by assessing colony formation on a serial dilution of ampicillin on agar plates (Supplementary Fig. S7A and Fig. 5) and growth on agar plates containing MIC strips (Liofilchem, Supplementary Fig. S7B and Fig. 5).

Intrigued that nearly all designs (11 of 14) were able to confer resistance to ampicillin in bacteria, we next investigated the biochemistry of the enzymatic reaction. Each design was expressed in *E. coli* and purified. The catalytic activities on the colorimetric β-lactam substrate nitrocefin were measured and initial velocities (Supplementary Fig. S8) were fit to the Michaelis-Menten equation (Methods, Fig. 3C, Supplementary Fig. S9, and Supplementary Table S1). All of the designs that enabled resistance to ampicillin in the biological assays had a similar catalytic efficiency ($k_{cat}/K_M$) to WT TEM-1. This includes the 70.a design with 84 amino acid differences relative to any known protein. The remaining designs that did not confer resistance to ampicillin in bacteria had no detectable biochemical activity (70.b), or were unable to be purified (50.a, 50.b). Although several designs had an increase in $k_{cat}$ (98.b, 90.b, opt.a, opt.b), the overall effect of this increase on catalytic efficiency was nullified by a concordant increase in $K_M$ for 90.b and opt.b. Many of the designs also appeared to show substrate inhibition at the maximum tested concentration of nitrocefin (Supplementary Fig. S9). In summary, these results offer a biochemical explanation (β-lactam hydrolysis) for each designs' ability to confer ampicillin resistance in bacteria.

We also performed biochemical analysis of each designs' activity on ampicillin. As absorbance-based detection of ampicillin cleavage was noisy in high-throughput plate-based formats and in low concentrations of ampicillin, we used a single concentration of ampicillin to determine initial velocities (Supplementary Fig. S10) and calculate specific activity for each design (Fig. 3B). All of the designs that enabled resistance to ampicillin in bacteria (except for opt.b) had activities similar to WT TEM-1, and the designs that did not confer resistance in bacteria had activity values similar to the catalytically dead negative control (neg. ctrl) or could not be purified (50.a and 50.b). The one exception to this agreement was opt.b, which, compared with the negative control, did not have different biochemical specific activity, but did have an increased MIC in the bacterial assays. Overall, as with the nitrocefin biochemical analysis, specific activity on ampicillin further confirms that the majority of designs are able to hydrolyze β-lactams.

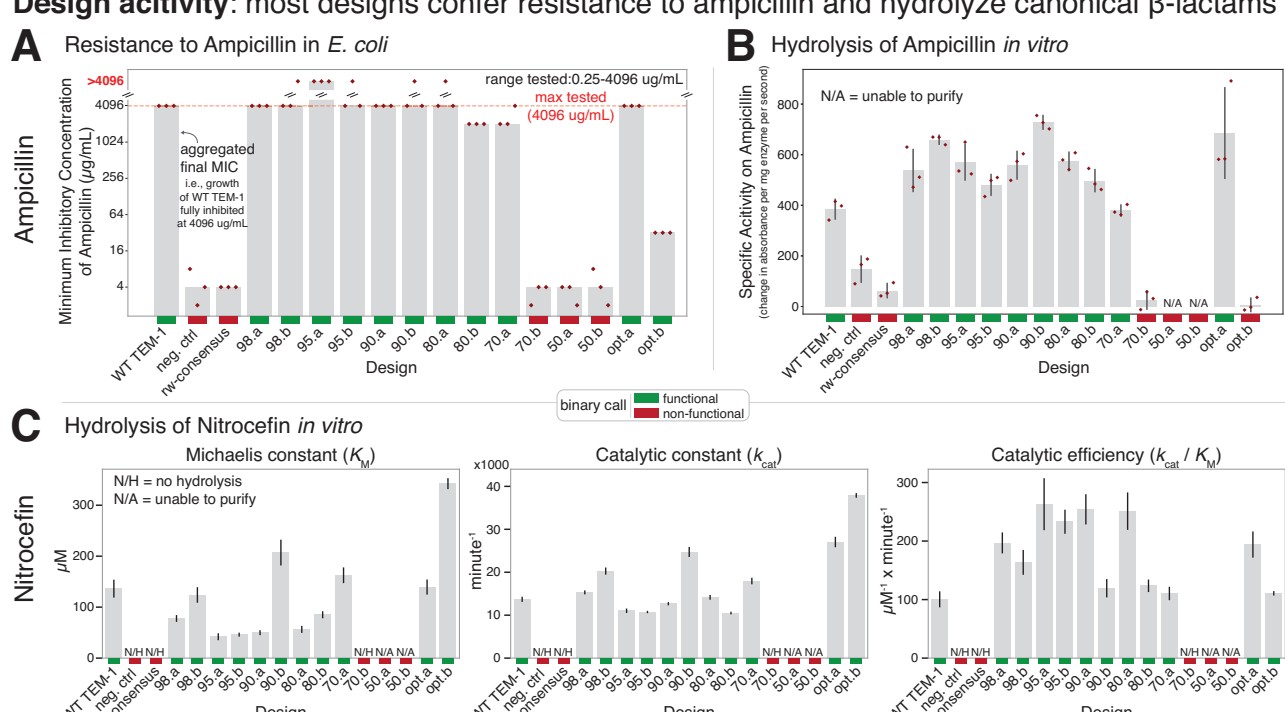

**Design acitivity**: most designs confer resistance to ampicillin and hydrolyze canonical β-lactams

**Fig. 3 | Most designs confer ampicillin resistance to bacteria as well as hydrolyze both ampicillin and nitrocefin in-vitro. A** Minimum inhibitory concentration (MIC) of ampicillin in *E. coli* as determined by a CLSI broth microdilution assay in which a fixed concentration of design-expressing bacteria were subjected to various ampicillin concentrations. The aggregated MIC calls (gray bars, see Methods) summarize three individual replicate experiments (red diamonds). Most designs (11 of 14) confer resistance to ampicillin. Three designs (70.b, 50.a, 50.b) have a similar MIC as the negative control (a catalytically dead point mutant). **B** Specific activity (change in absorbance per mg enzyme per second) of purified designs on the antibiotic ampicillin. Gray bars: mean of three replicate experiments (red diamonds). Error bars: standard deviation. Hydrolysis was measured at an absorbance of 235 nm with an initial concentration of 800 μM ampicillin. Specific activity values are generally consistent with the results of cell-based resistance experiments. Error bars: standard deviation. **C** Michaelis-Menten kinetics of each design towards the canonical colorimetric β-lactam substrate nitrocefin. Error bars: standard error of the model fit to all three replicate initial velocities on six or seven substrate concentrations (Methods). N/H: no hydrolysis was detectable (neg. ctrl, consensus, and 70.b). N/A: designs that could not be purified (50.a and 50.b). [colored bars] Binary call for each sample and assay. Red: non-functional. Green: functional. Source data are provided in the Source Data file.

Taken together, designs with the largest number of amino acid differences (50.a, 50.b, 70.b) were non-functional in both bacterial resistance assays and biochemical analysis. The other designs (11 out of 14) conferred resistance to ampicillin in bacteria using multiple independent assays and exhibited β-lactam hydrolysis in biochemical analysis of nitrocefin and/or ampicillin. These functional designs had varying numbers of amino acid changes, including two sequences with over 50 amino acid changes (80.a and 80.b), one with 62 changes (opt.b), and one with 84 changes (70.a) relative to any known homolog. These data are consistent with the general view that it is increasingly difficult to maintain or improve activity with an increasing number of amino acid changes in a protein, whether by computation design (this work) or experimental-based approaches[1]. The key encouraging difference, however, is that the design process used here can maintain function with a much larger number of mutations than a random mutation process.

**Designs have increased stability and increased activity on additional substrates**

Enhanced enzyme stability and altered substrate specificity or catalytic profile are common goals of protein design. The spectrum of protein stabilities and catalytic substrates of the β-lactamases included in the MSA used to derive the computational model likely extends far beyond that of WT TEM-1. As the design process is informed by this ensemble of homologous proteins with various characteristics (melting temperature, specificity, enzyme kinetics,

etc.), it is plausible that the designs are not just an optimized version of WT TEM-1, but rather that each reflects information from all of the sequences used in the MSA.

We assayed the melting temperature ($T_m$) of each purified design using differential scanning fluorimetry (DSF, Supplementary Fig. S11). Every design we were able to purify (all except 50.a and 50.b) had substantial increases in $T_m$ relative to WT TEM-1 (Fig. 4A). The WT TEM-1 $T_m$ is 50.6 °C and the absolute $T_m$ of each design ranged from 55 °C to 78 °C. Over half (9 of 14) of the designs had an increase of over 10 °C, and three exceeded a 20 °C increase. Additionally, these increases in thermostability were not at the expense of enzymatic activity as, aside from 70.b, all of these designs were also functional (at mesophilic temperatures). In general a protein's consensus sequence often has increased thermostability[38], and indeed the rw-consensus (reweighted consensus, Methods) of our MSA had a large $T_m$ increase of 15 °C. Although the computational fitness prediction takes into account positional conservation, all of the design sequences are substantially different from the rw-consensus (Fig. 2), suggesting that these increases in $T_m$ are not solely accounted for by conserved residues. In summary, every design that we were able to purify had increased thermostability, and, despite these increases, all of these designs except 70.b conferred resistance to ampicillin in *E. coli* and were able to hydrolyze nitrocefin (Fig. 3).

We next profiled the designs' ability to confer resistance to a panel of β-lactam antibiotic substrates using the same CLSI broth microdilution assay used to profile ampicillin resistance. Interestingly, many

**Property enhancements**: all designs have increases in thermostability and most have a broadening of available substrates.

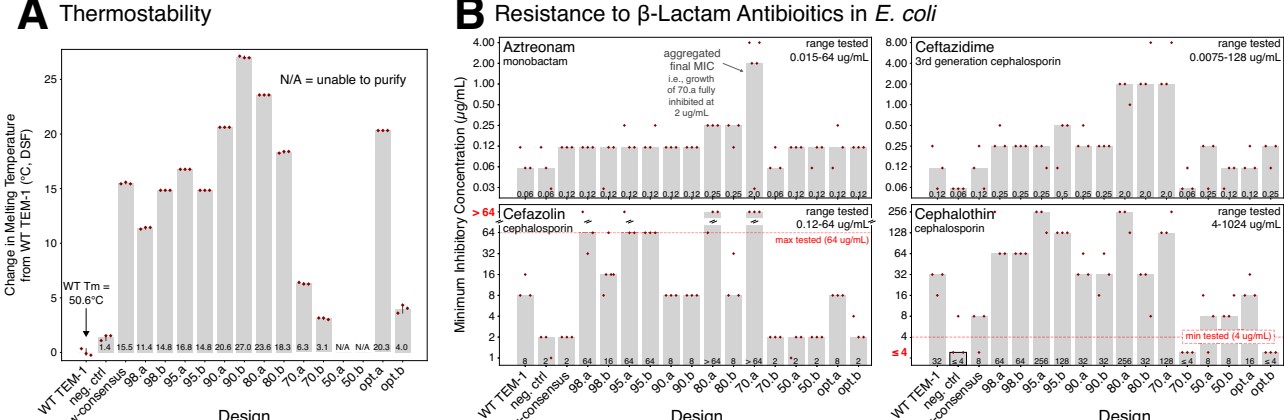

**Fig. 4 | All of the designs have increased stability and many also provide increased resistance to multiple classes of β-lactam antibiotics relative to WT TEM-1.** [**A**, Thermostability] Quantification of the change in thermostability ($T_m$) relative to WT TEM-1. $T_m$ was determined using differential scanning fluorimetry (DSF). Gray bars: mean of three replicate experiments (red diamonds). Error bars: standard deviation. Except for the two designs that could not be purified (50.a and 50.b), all designs were more thermostable, with 9 of the 11 functional designs having >10 °C increase in melting temperature. [**B**, Resistance to multiple β-lactams] Minimum inhibitory concentration (MIC) of several classes of β-lactam antibiotics was determined using CLSI broth microdilution. Gray bars: aggregated MIC call (Methods). Red diamonds: three individual replicate experiments. Upper left in each plot is the antibiotic with its class name. Many of the designs confer resistance to higher concentrations of antibiotics than WT TEM-1 (left-most column). Source data are provided in the Source Data file.

of the designs acquired increased activity on multiple substrates relative to WT TEM-1 (Figs. 4B and Fig. 5). Other than the two designs that were optimized solely for predicted fitness (opt.a and opt.b), most of the designs that were able to confer resistance to ampicillin also conferred increased resistance to *E. coli* grown in the presence of at least one of the other tested β-lactam antibiotics (aztreonam, ceftazidime, cefazolin, and cephalothin). Most notably, the 70.a design, which contains 88 mutations relative to WT TEM-1, had a ~32-fold increased MIC of the monobactam β-lactam antibiotic aztreonam, a 16-fold increased MIC of ceftazidime, an 8-fold increased MIC of cefazolin, and a 4-fold increased MIC of cephalothin. The 80.a and 80.b designs also had a 4-fold increased MIC of aztreonam, as well as a 16-fold increased MIC of ceftazidime. Many of the designs had an increased MIC of cefazolin (98.a, 95.a, 95.b, 80.a, and 70.a) and cephalothin (98.a, 98.b, 95.a, 95.b, 80.a, 70.a). Three of the antibiotics (cefoxitin, imipenem, and meropenem) had no consistent differences in MIC compared with the negative controls or WT TEM-1 (Supplementary Fig. S12). We obtained similar results when assessing resistance to aztreonam, ceftazidime, and cephalothin using MIC strips (Supplementary Fig. S13).

In summary, every design had an increase in thermostability, and most of the designs had an increase in the ability to confer resistance towards at least one of the tested β-lactams. In particular, the 80.a and 70.a designs had between 4-fold and 32-fold increased MIC of four tested antibiotics as well as 24 °C and 6 °C increased $T_m$, respectively. Most of the rest of the distance-constrained designs (98.a, 98.b, 95.a, 95.b, and 80.b), which all had more than an 11 °C increased melting temperature, also had increased activity on at least one of the four tested β-lactams. The simultaneous increase in thermostability and activity on some substrates suggests that the design process enhanced multiple parameters, resulting in a diverse set of designed variants that contain a set of useful properties.

### Highly mutated design variants have 3D protein structures nearly identical to WT TEM-1

To examine the structural effects of these mutations, we obtained X-ray crystal structures of designs 80.a, 80.b, and 70.a (Fig. 6A and

Supplementary Fig. S14). These three designs had some of the highest mutation counts relative to any natural sequence while still retaining function. Aligning these structures in 3D to a published WT TEM-1 structure (PDB: 1XPB[34]) revealed that all three have nearly identical Cα backbones to WT TEM-1 (0.26–0.61 Å RMSD over all Cα atoms, Fig. 6A). We also searched for local structural changes by examining differences in all pairwise distances between residues, which showed small changes in a one looped region for all three structures (positions 255–257) and in a second loop of 70.a (positions 53–55) (Supplementary Fig. S15, Supplementary Fig. S16, and Supplementary Fig. S17). Both 80.a and 80.b have 55 amino acid substitutions relative to WT TEM-1, of which 80.a has 15 mutations in the core of the structure and 80.b has 19 in the core (Fig. 6B). 70.a has 88 mutations relative to WT TEM-1, 35 of which are in the core.

Although all three designs had Cα backbone structures that were nearly identical to WT TEM-1, there were slight deviations, and so we analyzed structural variations among WT TEM-1 natural homologs. We collected a set of β-lactamase structures (947 polypeptide chains from 542 PDB: structures, Methods) and structurally aligned each as well as the 70.a, 80.a, and 80.b structures to WT TEM-1 (PDB: 1XPB). The Cα backbones of the homologous structures largely overlap with one another and those of the three designs (Fig. 6C). Quantifying the structural deviation from WT TEM-1 (PDB: 1XPB) reveals that the designs have a similar amount of variation as the natural homologous structures (Inset of Fig. 6C). We next investigated the relationship between sequence identity (from WT TEM-1) and structural variation, and found that the three designs have similar structural variation as other PDB entries that also have ~70–80% sequence identity from WT TEM-1 (Fig. 6D). In summary, the mutations introduced by the design process do not lead to larger structural variations than those of naturally evolved proteins, in spite of the imposed design constraint of upper bounds on the sequence distances not only to WT TEM-1, but also to all known natural homologs.

We next attempted to find structural evidence for the increased melting temperature and altered substrate activity profiles of the three functional designs for which we were able to obtain crystal

## Characterization summary: general agreement of design performance and properties across multiple independent assays

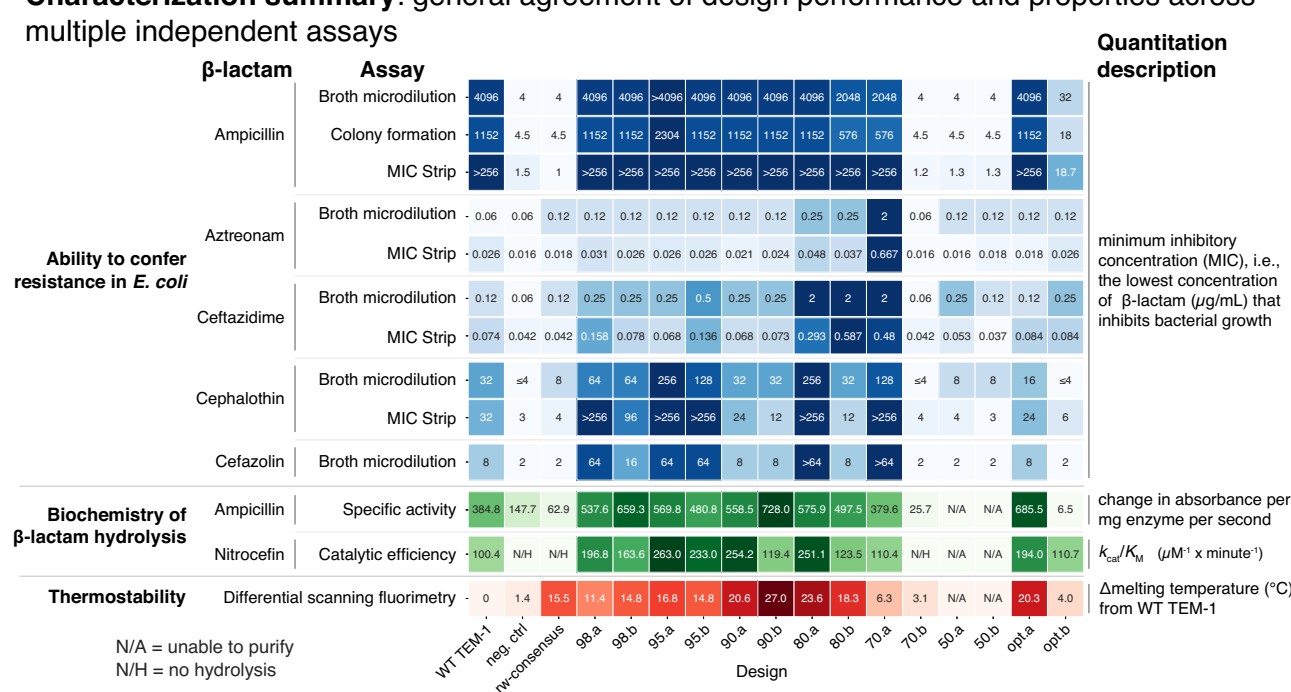

**Fig. 5 | Summary of in-cell, biochemical, and stability properties of designed variants.** [Blue color scheme]: Summary values from multiple independent resistance assays of the minimum inhibitory concentration (MIC) in *E. coli* of multiple *β*-lactams. Numbers for each assay are as follows: [broth microdilution] aggregate of three or five replicates (Methods), [colony formation] mean of three replicates, [MIC Strip] mean of three replicates unless any replicate was above the highest tested dose at which point the mode is displayed if available otherwise the median is depicted. Resistance assay colors are log normalized; darker colors: larger values. [Green and red color scheme]: in-vitro biochemical analysis of each design's ability to hydrolyze ampicillin (green, mean of three replicates) or nitrocefin (green, model parameters fit to three replicates) as well as the change in melting temperature (thermostability, mean of three replicates) from WT TEM-1 (red). N/A: the design could not be purified; N/H: the design had no hydrolysis. Biochemical colors are linear; darker colors: larger values. Source data are provided in the Source Data file.

structures. To investigate the structural basis for the increased $T_m$ (80.a had a 24 °C increase relative to WT TEM-1, 80.b had a 18 °C increase, and 70.a had a 6.3 °C increase, Figs. 4, 5), we tallied the total number of hydrogen bonds and the number of atom pairs in contact (1.7–4 Å) in each structure. These analyses did not reveal any substantial differences with WT TEM-1 that would explain the increase in $T_m$ (see Source Data, Structural Mechanism Analysis). However, we did observe literature-reported globally stabilizing mutations such as M182T in every one of the designed sequences, which has been shown to confer a 7.5 °C increase in $T_m$ as a single substitution[28]. The 70.a, 80.a and 80.b designs also showed increased resistance to aztreonam (80.a and 80.b had a 4-fold increased MIC, and 70.a had a 32-fold increase MIC, see broth microdilution assay in Figs. 4, 5). Analysis in and around the active site area of residue B-factors and solvent accessible surface area did not reveal any consistent differences between any of the design structures compared with the three PDB structures (1XPB, 4GKU, 1S0W[34,39,40]) that have the exact same sequence as WT TEM-1 (see Source Data, Structural Mechanism Analysis). There are five shared mutations near the active site that are in three designs (M69A, E104T, P167T, E168A, E240G) that may contribute to this increased resistance (Supplementary Fig. S18). Compared with published structures of *β*-lactamases bound to aztreonam (PDB: 5G18, 1FR6, 2ZQC, 4WBG, 4X53, 5KSC)[39,41–44], and penicillin-binding proteins (PDB: 3PBS, 3UE0, 5HLB, 6KGU)[45–48] the smaller side chains of E104T and E240G possibly avoid steric hindrance with aztreonam, but further experimental testing is necessary in order to arrive at a definitive biophysical explanation for the increased resistance.

In summary, the three designs possessed identical folds and highly conserved backbone conformations relative to WT TEM-1. The differences in structural variation were of similar magnitude to other published structures with a similar sequence identity to WT TEM-1 (~70–80%), and the mechanism for increased thermostability and increased activity on multiple substrates was not due to any obvious structural changes.

### The ensemble of mutations in each design positively influences fitness beyond that of individual point mutations

Thus far, this work indicates that our design algorithm, which utilizes a scoring function that takes into account both positional constraints and pairwise interactions between positions (i.e., epistasis between positions), can generate sequence variants with a very large number of amino acid changes (up to 84 in a single sequence) that maintained function and had enhanced properties. It is common to utilize experimentally-derived fitness measurements of individual point mutations to inform protein design. For example, deep mutational scans (DMS), which aim to quantify the fitness of all single point mutations in a wild type background, are a useful means to increase the yield of obtaining functionally active variants by avoiding the introduction of deleterious mutations[49,50]. However, the fitness effect of point mutations on the wild type sequence is not necessarily additive and likely becomes less useful for design as the variant sequences diverge substantially from wild type.

We examined the experimentally-determined fitness effect of individual amino acid changes in the functional designs in the WT TEM-1 background from a published DMS[25] (Supplementary Fig. S19). We defined "fitness defect" as an amino acid substitution having a fitness score of less than -1 in at least one replicate in the published DMS study, which conceptually equates to a 10-fold decrease in

**Structural analysis:** functional designs with highly divergent sequences have closely aligned backbone conformations to WT TEM-1 and natural β-lactamases.

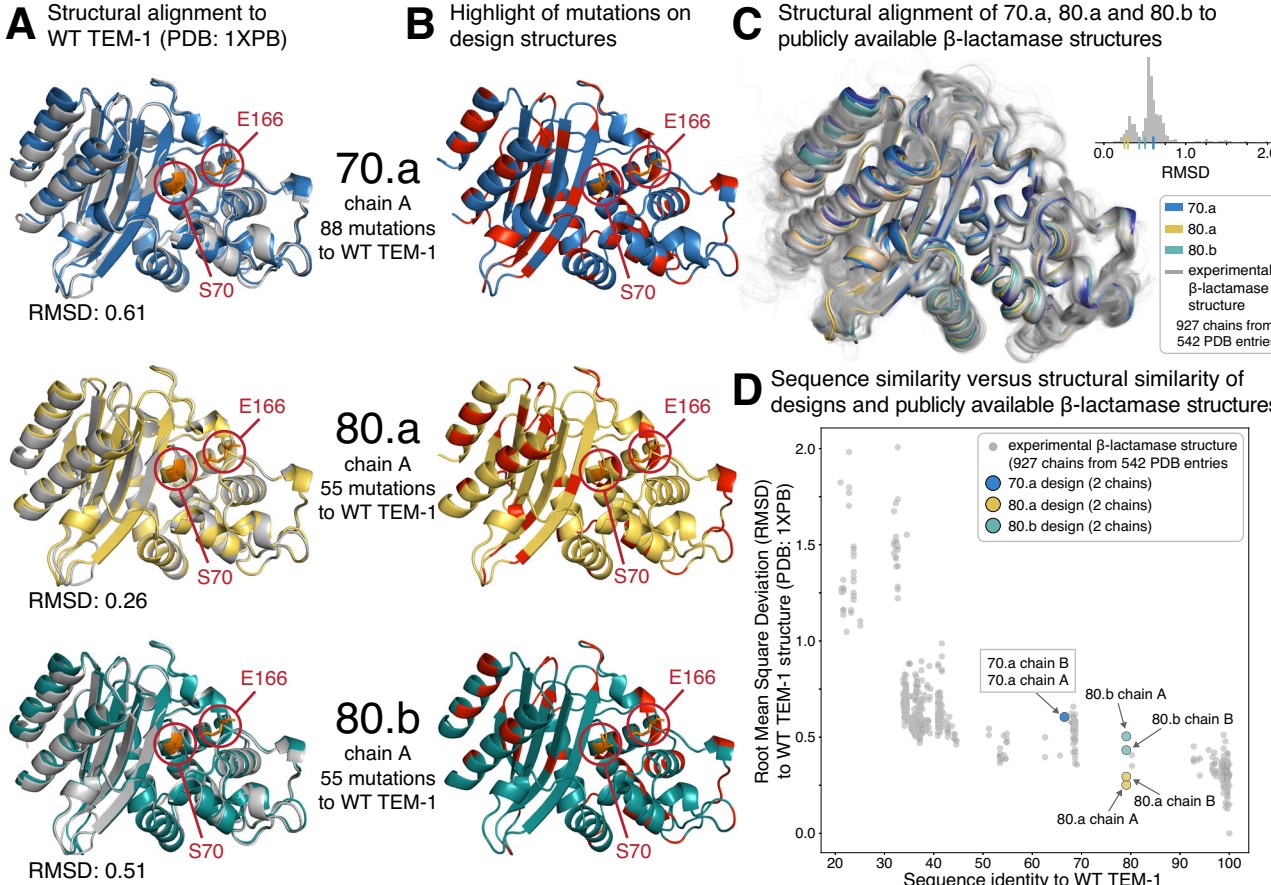

**Fig. 6 | Structural evaluation of designs. A** Structural alignment of the crystal structure of 70.a (top, blue), 80.a (middle, yellow) and 80.b (bottom, teal) with WT TEM-1 (PDB: 1XPB, silver). Catalytic residues E166 and S70[74] are circled in red and their side chains are in stick representation. **B** Highlight of the mutations (red sections of the ribbons) in each design relative to WT TEM-1. Catalytic residues (E166 and S70) marked as in [**A**]. **C** Structural superimposition of 70.a (blue), 80.a (yellow) and 80.b (teal) with publicly available β-lactamase structures (silver, 927 protein chains from 542 PDB entries). [**C**, inset] Distribution of structural similarity (root mean square deviation in Cα atomic positions, RMSD) of each publicly available β-lactamase structure relative to WT TEM-1 (PDB: 1XPB). RMSD of each design: marginal ticks on the x-axis. **D** The relationship between sequence identity and Cα backbone structural similarity (RMSD) for the same publicly-available β-lactamase structures as in [**C**] and the three designs. Colors as in [**C**]. Source data are provided in the Source Data file.

fitness at a high ampicillin concentration (2500 μg/mL) relative to WT TEM-1. Designs with the fewest number of amino acid changes (98.a, 98.b, 95.a, 95.b, 90.a, 90.b, opt.a) did not contain any mutations that exhibited a fitness defect at any concentration of ampicillin. However, the other functional designs contain amino acid changes that exhibit fitness defects in WT TEM-1; 80.a has two amino acid changes that show fitness defects, 80.b has four, 70.a has 11 and opt.b has 28. The presence of mutations that cause fitness defects to WT TEM-1 in four of the functional designs was noteworthy, especially given that all of these mutations also had a negative fitness prediction as point mutants in the WT TEM-1 background. Several key questions arise from these observations. How are these designs able to function with mutations that are deleterious to WT TEM-1? Why does the design generation algorithm, which attempts to optimize the predicted fitness, produce designs with mutations that were predicted to have a negative fitness impact to WT TEM-1? We next investigated these questions by (1) performing a targeted analysis on one particularly well studied inactivating mutation (G251W), and (2) taking a more general look at the predicted fitness effects of a set of mutations known to be deleterious to WT TEM-1 in each design.

The G251W mutation, which was present in both 70.a and opt.b, stood out due to multiple studies that describe it as negatively impacting fitness in WT TEM-1[25,51]. In addition, the G251W mutation in WT TEM-1 had one of the lowest predicted fitnesses of any point mutation in the functional designs. The opt.b design had the lowest performance of any functional design in both the bacterial resistance and biochemical assays, which could in part be due to G251W. Intriguingly, 70.a was reasonably active on ampicillin and was the most effective design (highest MIC) towards three of tested β-lactam antibiotics (Figs. 2, 4, 5).

We next investigated whether any single mutation in the 70.a design enables it to retain function with the presence of G251W. In the literature[51], several "compensatory mutations" have been described to at least partially alleviate the fitness defects caused by G251W, and both 70.a and opt.b contain some of these mutations. However, each of these mutations was also present in at least one other design that did not contain the G251W mutation, suggesting that these mutations were not specifically selected by the design algorithm to compensate for G251W. It is also possible that the design generation algorithm selected compensatory mutations not uncovered in the literature review. Although there are no large predicted epistatic effects between G251W

**Context matters**: the set of mutations in each design collectively influence the predicted effect of individual point mutations.

**A** Design point mutations are generally predicted to have a negative fitness effect in WT TEM-1 and a positive effect in each design

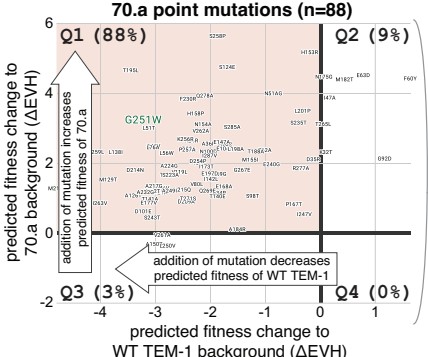

**B** 70.a mutations introduced into WT TEM-1 and reverting mutations in 70.a back to WT TEM-1 are both predicted to negatively effect fitness

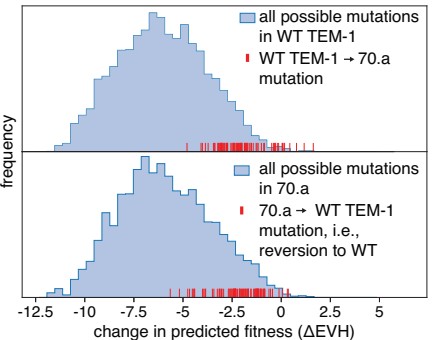

| design variant | mutation count to WT TEM-1 | percent of mutations | | | |
|---|---|---|---|---|---|
| | | Q1 | Q2 | Q3 | Q4 |
| 98.a | 10 | 30 | 70 | 0 | 0 |
| 98.b | 10 | 10 | 90 | 0 | 0 |
| 95.a | 17 | 35 | 59 | 6 | 0 |
| 95.b | 17 | 35 | 59 | 6 | 0 |
| 90.a | 30 | 70 | 30 | 0 | 0 |
| 90.b | 30 | 57 | 33 | 10 | 0 |
| 80.a | 55 | 73 | 18 | 9 | 0 |
| 80.b | 55 | 78 | 18 | 4 | 0 |
| 70.a | 88 | 88 | 9 | 3 | 0 |
| 70.b | 88 | 88 | 9 | 3 | 0 |
| 50.a | 131 | 85 | 4 | 11 | 0 |
| 50.b | 130 | 86 | 3 | 11 | 0 |
| opt.a | 31 | 61 | 39 | 0 | 0 |
| opt.b | 144 | 97 | 3 | 0 | 0 |

**Fig. 7 | The collective influence of the set of mutations in each design on the predicted effect of individual point mutations.** [**A**, left] Example design (70.a) of the raw data that are summarized in the table on the right. Plotted is a comparison of how the addition of each point mutation in 70.a changes the predicted fitness (ΔEVH) in WT TEM-1 (x-axis) versus 70.a (y-axis). Each of the 88 points represent a single amino acid substitution that is found in 70.a. Pink background indicates the top left quadrant, which are those mutations that are predicted to have a negative effect on fitness in WT TEM-1 and positive effect on fitness in 70.a. The percent of amino acid changes in each quadrant is indicated. [**A**, right] Summary of predicted fitness changes (ΔEVH) for each designs' amino acid changes compared with their predicted effect in WT TEM-1. The "percent of mutations" is the percent of mutations in each quadrant (see example on left), with the pink background containing the percentage that are predicted to have a negative fitness effect in WT TEM-1 and positive effect in the design. **B** Distribution of predicted fitness effects (ΔEVH) of all possible amino acid changes (252 * 19) introduced in the WT TEM-1 sequence background (top, blue) and 70.a sequence background (bottom, blue). Red lines are the predicted fitness effect of the 70.a mutations if added individually to WT TEM-1 (top) or removed from 70.a (i.e., reverted to the WT TEM-1 amino acid). Source data are provided in the Source Data file.

and the 70.a mutations on the WT TEM-1 sequence background (Supplementary Fig. S20), these double mutants do have some of the highest predicted changes in fitness of all possible double mutations (Supplementary Fig. S21). In the 70.a structure, new local interactions between mutated residues W251 and R230 and non-mutated residue E212 may contribute to maintaining β-lactamase structure and function (Supplementary Fig. S22). In summary, (1) 70.a contains many point mutations that have been described in the literature as compensatory, (2) these mutations combined with G251W have some of the highest predicted fitness effects, and (3) structural analysis suggests that local interactions between W251 and R230 (a mutation in 70.a F230R) may be important for maintaining function. Each or all of these observations may explain how 70.a is able to overcome the deleterious effect of the G251W mutation.

Rather than explaining the presence of mutations in the designs that would have caused fitness defects in WT TEM-1 solely through a single compensatory amino acid change, it is plausible that it is the ensemble of mutations in a given design that accounts for why the algorithm selects such mutations. To probe this hypothesis, we examined how the predicted fitness score of a point mutation may be effected by the amino acids in the rest of the sequence (i.e., the "background" for the point mutation). Using the 70.a design as an example, we isolated each of its 88 mutations and calculated the change in predicted fitness (ΔEVH) from the WT TEM-1 amino acid to the 70.a amino acid in both the WT TEM-1 background and 70.a background (Fig. 7A, left). The predicted fitness effect of each point mutation is positive (+) or negative (-) to 70.a and/or to WT TEM-1, and therefore each mutation falls into one of four possible categories: (Q1) positive in 70.a and negative in WT TEM-1, (Q2) positive in both 70.a and WT TEM-1, (Q3) negative in 70.a and positive in WT TEM-1, and (Q4) negative in both 70.a and WT TEM-1. For all designs, the percent of mutations in each category is shown in Fig. 7A (right). As expected, nearly all of the mutations are predicted to have a positive effect on fitness in the design backgrounds. Interestingly, a large fraction of the design mutations are predicted to have a negative impact in the WT

TEM-1 background. For example, G251W in WT TEM-1 is predicted to have an ~3 point decrease (arbitrary units) in fitness whereas the same mutation in 70.a is predicted to have a ~3 point increase (Fig. 7A, left). Conceptually, even though introducing an individual mutation from the designs into WT TEM-1 often lowers its predicted fitness, adding the same individual mutation to the set of other design mutations leads to a higher predicted fitness. In the context of the designed sequence, reverting the mutated residues individually to their WT TEM-1 amino acids lowers the design's predicted fitness (Fig. 7B). These data suggest that the specific sequence context is very important, i.e., that the combination of all mutations can drastically influence the fitness effect of individual point mutations. Furthermore, the site and pairwise evolutionary model is able to capture complex epistatic interactions between amino acids and, with reasonable accuracy, correctly predict the fitness of highly mutated designs.

## Discussion

In this work we demonstrate a design algorithm based on an evolutionary model of sequence co-variation that enables large changes to primary sequence while enhancing function and thermostability. Previous studies have demonstrated that the random sequential introduction of mutations into β-lactamase results in a rapid decay of activity (resistance to ampicillin), resulting in complete ablation of activity in nearly every variant after the introduction of only 10 mutations[1]. Of the small set of 14 β-lactamase designs that we tested, the hit rate of active sequences, with up to 30% of residues mutated, was surprisingly high with 11 designed proteins enabling bacterial growth on ampicillin and hydrolysis of nitrocefin. These functional designs contained between 7 and 84 amino acid differences from any known protein, indicating that our algorithm far outperforms random mutagenesis-based approaches.

Perhaps one of the most interesting results is the joint optimization of both stability and activity, which has often been viewed as an inherent tradeoff in the protein engineering literature[52–54]. In addition to maintaining enzymatic activity, all of the functional designs also had

thermostability increases of between 6 °C and 27 °C, and most conferred increased resistance to one or more β-lactam antibiotics. Stability has been shown by multiple groups to correlate with the predicted fitness from evolutionary models (e.g., Potts-like models, including EVcouplings and direct coupling analysis)[10,13,14], and we now show that this stability can extend to synthetic sequences with a high mutation count. Crystal structures of the functional designs with the highest mutation count to any natural sequence revealed nearly complete structural preservation relative to WT TEM-1 β-lactamase. The increase in stability and resistance to β-lactams plausibly emerges from the information in the collective set of homologs, which evolved in many species over millions of years, used in sequence fitness prediction. Individual sequences only need to be stable enough to function in their native conditions, and therefore likely only require a subset of the many possible molecular mechanisms of stabilization. One simple explanation for why the designed variants (and often consensus sequences in general) exhibit increased stability is that stabilizing amino acids and interactions are likely over-represented in the naturally occurring proteins in the multiple sequence alignment used for model inference. The expansion of substrate specificity in some of the designs is plausibly related to the diversity of substrate preferences in different species and different conditions represented in the set of sequences in the multiple sequence alignment.

We expect this design strategy leveraging natural diversity to engineer stability and activity is broadly applicable to a range of protein families. The β-lactamase family in particular has a high level of functional and sequence diversity, as reported, with over 4000 known enzymes in 17 functional groups differentially targeting four classes of substrates[55,56]-characteristics that likely facilitate learning of a fairly general framework of constraints that retain fold and function while allowing for a range of substrate specificities. A previous study using a similar modeling framework demonstrated the generation of a large set of functional chorismate mutase variants of considerable sequence diversity, illustrating the potential broad applicability of this type of design process[2]. The addition of complementary data sources may further enhance the general utility of the method, especially for proteins that lack rich evolutionary sequence information. For example, structural information could be used to prioritize known interactions in the objective function or filter out structurally unlikely candidate designs[4]. It remains to be determined what levels of sequence and functional diversity in the training alignment are necessary for designing libraries of stable variants with increased activity, and which protein families satisfy these requirements.

It is tempting to conclude that these designs, which have large numbers of primary sequence changes while maintaining or even increasing activity, are better starting points for refining the specificity of a design in new directions, e.g., by further exploration of very similar sequences in the neighborhood of the designed starting point, rather than the original starting sequence. One common approach for refining the properties of a protein is directed evolution where mutations in a starting sequence are gradually introduced and accumulated (usually in a greedy manner) over multiple rounds of selection for the desired properties. As the negative effect of mutations can often be linked to decreased stability[57], starting with a protein that has high stability is useful for directed evolution as increased mutational tolerance enables a higher "hit rate" of stable sequences. All of the purifiable designs had increased stability with over half having melting temperature increases of more than 10 °C from the wild type. This increased stability suggests that the majority of the designed sequences may be more tolerant to mutation than WT TEM-1, which would be useful in traditional protein optimization strategies like directed evolution. Conceptually, the design process presented here enables large "jumps" to new regions of functional sequence space through the introduction of many

mutations, and further improvements to optimize specific functional and structural properties could then be achieved through smaller mutational steps, i.e., "walking" in sequence space. We believe this "jump and walk" strategy may enable us to efficiently discover protein variants with diverse structural and functional property changes while maintaining or increasing function.

One interesting methodological question that arises from these results is how important accounting for epistasis is for our design strategy, and to what extent other machine learning models[5,7,18,58-61] that explicitly include higher-order dependencies, such as variational auto-encoders and large-scale language models, are also able to capture the collective effects of protein stability and function necessary to perform a similar design strategy. The EVcouplings or Potts model[10] used in this work does approximate collective effects by inferring coupling terms (residue-residue interactions) up to second order (pairwise) that best describes the full dataset of available sequences. When these interactions act iteratively through the entire system, collective effects are approximately captured, in analogy to similar, highly successful, models in statistical physics. Similarly, methods such as variational auto-encoders are also an approximation with parameters derived by minimizing the difference between encoded (input) and decoded (generated) sequence distributions. In practice, which approximation best captures collective effects for the design of entirely new sequences depends on the particular problem and remains to be determined in each case or against large carefully crafted benchmark datasets.

Related work has recently reported[62] the generation of WT TEM-1 variants using similar models together with an evolution-like variant sampling and selection algorithm. Some of the variants generated by this computational design process contained many mutations, and the authors report that some of the variants are more active than wild type. In separate reports, the same group used similar models that evaluate co-evolutionary patterns to create variants with a small number of mutations that modulate DNA-protein and RNA-protein binding interactions[63,64], highlighting how these models can also be used for molecular engineering beyond single-protein design. While different from what is presented here (both in goals and implementation details), collectively their work and ours demonstrate the general capability and utility of using co-evolutionary models for protein design.

This work supports the use of statistical models of evolutionary sequence information for protein design, enabling the simultaneous introduction of many mutations into the primary protein sequence while maintaining function. Future work will investigate the biophysical explanation for the enhanced properties obtained here, as well as how generalizable this enhancement strategy is to other proteins. We anticipate that this type of approach will be readily applicable to many protein classes as a means to enhance and design new industrial or therapeutic functions.

## Methods
### Design process and parameters
**Alignment.** A multiple sequence alignment of β-lactamases was constructed using five iterations of jackhmmer search (version 3.2.1) against the UniRef100 database with a length-normalized bitscore of 0.5 (selected to ensure primarily β-lactamases in the alignment). The alignment was filtered to exclude positions with more than 30% gaps and to exclude sequence fragments aligning to less than 50% of the target sequence. To account for redundancy, similar sequences in the alignment are re-weighted according to their uniqueness using a Hamming distance cutoff of 0.2[10].

**EVcouplings model.** The site and coupling parameters of the EVcouplings model were learned via regularized maximum

pseudolikelihood[8,10]. The model file is provided as Supplementary Data, which can be loaded and queried in python using the EVcouplings framework (https://github.com/debbiemarkslab/evcouplings).

**Monte Carlo sampling with diversity restraints.** Sequences were sampled via batch-Gibbs sampling on a penalized energy function derived from the EVcouplings Potts model. Batch Gibbs sampling produces a batch of sequences in by iteratively resampling random positions in random batch members according to a batch energy function. We anneal the inverse temperature on a linear schedule from 0.5 to 10 over 1000 batch sweeps.

The batch energy function can enforce batch-level constraints such as inter-sequence diversity. We defined the un-normalized joint energy function as

$$U(\boldsymbol{\sigma}_1, \ldots, \boldsymbol{\sigma}_M) = \sum_{a=1}^{M} U(\boldsymbol{\sigma}_a) \tag{1}$$

where $\{\boldsymbol{\sigma}_1, \ldots, \boldsymbol{\sigma}_M\}$ is a batch of $M$ sequences to be sampled. This batch objective can be factorized into per-sequence objectives $U(\boldsymbol{\sigma}_a)$ as a

$$U(\boldsymbol{\sigma}_a) = \underbrace{-\sum_{i=1}^{N} h_i(\sigma_{a,i}) - \sum_{i<j} J_{ij}(\sigma_{a,i}, \sigma_{a,j})}_{\text{Potts energy}} \tag{2}$$

$$+ \underbrace{\lambda_{\text{Target}} \mathbb{I}[D(\boldsymbol{\sigma}_a, \boldsymbol{\sigma}_{\text{target}}) \notin [d_{\min}, d_{\max}]] \left| D(\boldsymbol{\sigma}_a, \boldsymbol{\sigma}_{\text{target}}) - d_{\min} \right|}_{\text{Target penalty}} \tag{3}$$

$$+ \underbrace{\lambda_{\text{Diversity}} \sum_{b \neq a} \mathbb{I}\left[ D(\boldsymbol{\sigma}_a, \boldsymbol{\sigma}_b) < d_{\text{Diversity}} \right]}_{\text{Diversity penalty}} \tag{4}$$

$$+ \underbrace{\lambda_{\text{Alignment}} \sum_{b=1}^{N} \mathbb{I}\left[ D(\boldsymbol{\sigma}_a, \boldsymbol{s}_b) < d_{\text{Alignment}} \right]}_{\text{Alignment penalty}}. \tag{5}$$

where $h_i(\sigma_{a,i})$ and $J_{ij}(\sigma_{a,i}, \sigma_{a,j})$ are the fields and couplings of the EVcouplings model, $D(\boldsymbol{\sigma}_a, \boldsymbol{\sigma}_b) \in [0,1]$ is the normalized Hamming distance between sequences $\boldsymbol{\sigma}_a$ and $\boldsymbol{\sigma}_b$, and $\{s_1, \ldots, s_N\}$ is a reference multiple alignment of $N$ sequences.

We included three penalty scores in the batch objective: (i) a target distance score that penalizes sequences to be within a specified range of distances from the reference sequence, (ii) a batch diversity score that penalizes pairwise sequence similarity within a batch, and (iii) an alignment distance score that penalizes similarity to any known sequences in the multiple sequence alignment. Throughout our experiments, we used hyperparameters $\lambda_{\text{Target}} = 1000$, $\lambda_{\text{Diversity}} = \lambda_{\text{Alignment}} = 10$, and varied $d_{\text{Diversity}} = d_{\text{Alignment}} = d_{\min}$ to the satisfy the desired similarity to target. We set $d_{\max} = d_{\min} 0.05$ to add a small margin of tolerated target distances. Sampling six sequences with 1000 sweeps of batch Gibbs sampling took ~1 h on a 2018 2.2 GHz Intel Core i7 with a naive MATLAB (9.4 R2018a) implementation. We note that this could be greatly accelerated by modern gradient-based discrete sampling methods[65].

**Greedy sampling—opt.a.** One sequence (opt.a) was generated via a simple greedy sampling protocol in which the wild type sequence was iteratively mutated by the single most favorable point mutation in the one-mutant neighborhood until reaching a local optimum.

**Parallel tempering sampling—opt.b.** One sequence (opt.b) was generated via annealed parallel tempering for global optimization. Parallel tempering combines sampling processes across several temperatures that allows higher temperature exploration to inform lower temperature exploitation, and can be useful on rugged landscapes[35]. We used 19 replicas with inverse temperatures initialized with linear

spacing between 0.1 and 1.0. We increased these inverse temperatures over 1000 Gibbs sweeps by schedule $\beta_0 1.002^i$, where $i$ is the numbers of sweeps.

**Frequency reweighted sequence—rw-consensus.** The reweighted consensus control was generated by assigning the most frequent residue at each position in the alignment, after redundancy-reweighting each sequence according to its uniqueness with a Hamming distance cutoff of 0.2[10].

**Similarity to known sequences.** We used the BLAST to find the nearest homologs when preparing for publication. The database used was the non-redundant protein sequences (nr) and the algorithm selected was blastp (protein-protein BLAST, version 2.14.1). The closest natural homologs were only partially aligned for rw-consensus (253 out of 263 positions aligned) and opt.b (257 out of 263 positions aligned). The percent identity and number of mutations reported for these two sequences was based only on the aligned positions.

### Cloning for antibiotic resistance assays
**Plasmid preparation.** Designs were cloned into a modified pSTC0 plasmid after a native ampR promoter and WT TEM-1 N-terminal signal peptide. pSTC0 was a gift from Alfonso Jaramillo (Addgene plasmid #39240; http://n2t.net/addgene:39240; RRID:Addgene_39240)[66]. The pSTC0 plasmid originally contained two antibiotic cassettes, ampicillin and kanamycin. We replaced the kanamycin resistance cassette with a Zeocin resistance cassette in order to reduce the probability of contamination with other ongoing projects in the lab.

**Codon optimization.** Reverse-translation of sequences to DNA was performed using the canonical *E. coli* codon table. To reduce the impact of differential translation efficiencies on the rate of β-lactamase translation, we used codons for the mutant amino acid with the most similar codon usage frequency to that of the wild type codon.

**Gene synthesis and assembly into plasmid.** Designs were synthesized as gBlocks by Integrated DNA Technologies and Gibson cloned (New England Biolabs) into the modified pSTC0 backbone.

### Determination of bacterial resistance to ampicillin and other β-lactam antibiotics
Sequence-validated glycerol stocks of DH5α *E. coli* (New England Biolabs) were used for all bacterial resistance assays.

**MIC determination using a broth microdilution assay.** A fixed concentration of design-expressing *E. coli* (DH5α), as determined by optical density (OD) calibrated to a McFarland prep, was added to a 2-fold serial dilution of ampicillin in a cation-adjusted Mueller-Hinton broth. Three experiments were performed for each design-ampicillin concentration, and the final MIC was determined as (1) the mode of the 3 replicates or, if there is no mode, then (2) use the median if all replicates are in essential agreement (i.e., within one serial dilution of one another) or, if there is no essential agreement, then (3) perform an additional 2 replicates and use the median of 5 replicates.

**Ampicillin MIC determination by assessing colony formation on agar plates.** *E. coli* (DH5α) that expressed each design were used to inoculate Mueller-Hinton (MH) broth containing 50 µg/mL zeocin (to ensure plasmid maintenance). Cultures were grown overnight (37 °C at 250 RPM). The following day, cells were spun down and resuspended in 0.85% NaCl, and diluted to a final OD of 0.125. This dilution was further diluted to achieve 5000 cells/mL and 40 µL of this final solution (~200 cells) was pipetted into one well of a 6-well plate that contained a serial dilution of ampicillin in MH agar. For each design, two

6-well plates were used for each of the three replicates, which contained ampicillin at the following concentrations: 4608 µg/mL, 2304 µg/mL, 1152 µg/mL, 576 µg/mL, 288 µg/mL, 144 µg/mL, 72 µg/mL, 36 µg/mL, 18 µg/mL, 9 µg/mL, 4.5 µg/mL, 0 µg/mL. MIC was defined as the lowest ampicillin concentration having no visible colonies after overnight culture at 37 °C.

**MIC determination using MIC strips.** *E. coli* (DH5α) that expressed each design were used to inoculate Mueller-Hinton (MH) broth containing 50 µg/mL zeocin (to ensure plasmid maintenance). Cultures were grown overnight (37 °C at 250 RPM). The following day, cells were spun down and resuspended in 0.85% NaCl, and diluted to a final OD of 0.125. 500 µL of each design-expressing *E. coli* culture was pipetted onto three 15-cm plates containing MH agar+50 µg/mL zeocin, and 25 glass beads were dropped onto the plate and shaken vigorously until the liquid culture was well distributed. Five MIC strips (Ampicillin, aztreonam, ceftazidime, cefazolin, and cephalothin) were placed on the plate in a star pattern. Plates were placed at 37 °C overnight, and the following day plates were imaged with a Bio-Rad ChemiDoc. Each sample was quantitated by eye by assessing the intersection of bacterial growth and non-growth on the strip.

## Protein purification

All TEM-1 designs were expressed and purified as follows. Designed variants were cloned into the pCDB179 plasmid backbone, containing a cleavable N-terminal His-SUMO tag[67]. DNA was transformed into Lemo21(DE3) cells and grown in MDAG-135 media overnight at 37 °C. The saturated culture was used to induce an autoinduction media culture of TBM-5052 supplemented with 1 mM Rhamnose and Y-Antifoam (Sigma, A5758) as a 1:100 dilution. The expression culture was grown for 20–22 h at 30 °C. Cells were harvested by centrifugation and resuspended in Buffer A (20 mM NaPi, 500 mM NaCl pH 7.4, 20 mM Imidazole) supplemented with 0.25 mg/ml Lysozyme (Sigma, L6876), Turbonuclease (Accelagen, N0103), and EDTA-free Protease Inhibitor tablets. Cells were lysed by sonication at 4 °C and clarified by centrifugation at 14,000 x g for 20 min. The supernatant was applied to 2 mL of cOmplete Ni²⁺-agarose (Roche) prewashed with Buffer B (20 mM NaPi, 500 mM NaCl pH 7.4, 20 mM Imidazole) and batch bound at room temperature for 1 h. The Ni²⁺-agarose resin was washed with 60 mL of Buffer B, and sample was eluted with 5 mL Buffer C (20 mM NaPi, 500 mM NaCl pH 7.4, 400 mM Imidazole). Protein was concentrated to 0.5 mL using Amicon-4 10K concentrators and applied to a Superdex 75 pg 10/300 column equilibrated with Buffer D (20 mM NaPi, 150 mM NaCl pH 7.4). The β-lactamase peak was collected, and incubated with a 1:100 (w/w) dilution of His-CthSUMO protease[67] at room temperature for 1 h. The protein solution was incubated with 2.5 mL of Ni²⁺-agarose prewashed with Buffer D, and the flow through was collected as the final product. Concentration was measured by A280 and purity was determined by SDS-PAGE. Protein was brought to 5% (v/v) glycerol, aliquoted, flash-frozen in liquid nitrogen, and stored at −80 °C.

## Enzyme kinetics

All assays were performed at 25 °C in 20 mM NaPi, 150 mM NaCl pH 7.4. A SpectraMax i3x instrument was used to monitor substrate hydrolysis at wavelengths 482 nm for nitrocefin and 235 nm for ampicillin (delta E 900).

**Nitrocefin.** Purified designs and control enzymes (1 nM) were incubated with multiple concentrations of nitrocefin (12.5 µM, 25 µM, 50 µM, 100 µM, 200 µM, 400 µM, 800 µM), and hydrolysis was measured at an absorbance of 482 nm over time. For each plate, the buffer-only well absorbance was subtracted from each datapoint. A standard curve of absorbance versus hydrolyzed nitrocefin concentration was generated, and a linear fit of these data was used to calculate an absolute concentration of hydrolyzed nitrocefin for each sample. For

each sequence and nitrocefin concentration replicate condition, a linear regression was calculated for all possible sliding windows of four timepoints (Supplementary Fig. S8). The maximum slope was selected and the quality of fit was verified by normalized RMSD as well as by eye.

To calculate the $k_{cat}$ and $K_m$ for each enzyme, data were fit using the *lmfit* Python library to the Michaelis-Menten equation:

$$V = k_{cat}*E*S/(K_m + S) \qquad (6)$$

where $V$ = initial reaction rate ,
$E$ = enzyme concentration ,
$S$ = substrate concentration

Data were first fit to a maximum substrate concentration of 400 µM as we observed inhibition at 800 µM for some designs (Supplementary Fig. S9). If the resulting $K_m$ multiplied by three was more than the maximum substrate concentration of 400 µM (an indication that the analysis was not reliable) and there was no clear substrate inhibition at 800 µM, then data were then re-fit to the maximum measured substrate concentration of 800 µM. Designs fit to the 800 µM substrate maximum were WT TEM-1, 98.b, 90.b, opt.b. All other designs were fit to a maximum of 400 µM substrate. The equation fit compared to measured data (initial rate versus nitrocefin concentration) are shown in Supplementary Fig. S9. Kinetic parameters are plotted in Fig. 3C and are listed in Supplementary Table S1. Reported errors are the standard error directly reported from the *lmfit* python library and, for the $k_{cat}/K_m$, error propagation was applied to derive the standard error:

$$\sigma_{k_{cat}/K_m} = |k_{cat}/K_m|\sqrt{(\sigma_{k_{cat}}/k_{cat})^2 + (\sigma_{K_m}/K_m)^2} \qquad (7)$$

Samples neg. ctrl (S70A), rw-consensus, and 70.b had no detectable hydrolysis (N/H) and samples 50.a and 50.b were unable to be purified (N/A).

**Ampicillin.** Specific activity measurements were determined using a single initial concentration of ampicillin (800 µM), and a constant amount of enzyme (0.5 nM) for each enzyme. Hydrolysis was measured at an absorbance of 235 nm over time. For each replicate, a linear regression was calculated for all possible sliding windows of five timepoints (Supplementary Fig. S10). Initial reaction rate was determined as the negative of the steepest slope in which predicted values from the linear regression were consistent with the data (normalized RMSD < 0.001) so that the initial rate is related to product formation instead of substrate consumption as is measured by absorbance. Final specific activity was calculated by dividing the initial rate by enzyme concentration.

## Differential scanning fluorimetry (DSF)

Protein thermal stability was measured by differential scanning fluorimetry using a QuantStudio Pro 6/7. Protein was brought to a concentration of 10 µM with 5x Sypro Orange dye in a final volume of 20 µL in 20 mM NaPi, 150 mM NaCl pH 7.4. Raw data processing and curve fitting was determined using Applied Biosystems Protein Thermal Shift software, version 1.2, and the $T_m$ is reported as the "Tm D" value. Measurements were conducted in triplicate.

## X-ray structure determination

**Expression and purification.** Designs 80.a and 80.b were purified as described in Protein Purification. This purification method did not yield crystals for 70.a so a different strategy was applied. A construct with the sequence corresponding to the designed variant was synthesized by Genscript. The cDNA was inserted into a pET15b vector at the NdeI/BamH1 site, with an N-terminal His tag followed by a

thrombin protease recognition sequence. The DNA construct was transformed into BL21(DE3) cells and expressed in liter amounts in 2xYT media (FORMEDIUM, UK) in the presence of 100 µg/mL of ampicillin (SIGMA Co). Cells were grown at 37 °C to an OD600 of 0.6, induced with 0.1 mM IPTG, and incubated overnight at 18 °C. Cells were then harvested and lysed by sonication in extraction buffer (300 mM NaCl, 20 mM imidazole, 10 mM $\beta$-mercaptoethanol, 20 mM Tris-Cl pH 8). Following centrifugation (30 min, 4 °C, 12,000 × $g$), the soluble fraction was applied to a Ni$^{2+}$-agarose resin by gravity. The resin was washed in the extraction buffer supplemented with 40 mM imidazole, and bound proteins were eluted with a step gradient to 200 mM imidazole. Eluted proteins were dialyzed overnight in extraction buffer in the presence of thrombin protease at 4 °C. Following thrombin cleavage, there were 4 non-native residues (Gly-Ser-His-Met) prior to Pro27 of mature TEM-1 $\beta$-lactamase. Cleaved 70.a was collected as a flow-through fraction from a second Ni$^{2+}$-agarose gravity column. Following exchange into a low salt buffer (10 mM NaCl, 10 mM Tris-HCl pH 8, 1 mM DTT), the protein was loaded onto a Mono Q 5/50 GL ion exchange column and a 50% gradient of high salt buffer was applied (low salt buffer supplemented with 1 M NaCl). Fractions corresponding to the first peak of $\beta$-lactamase protein were pooled and loaded onto a Superdex 75 10/300 GF column (Cytiva) equilibrated in gel filtration buffer for further purification. The $\beta$-lactamase peak was concentrated in a 10 kDa molecular weight cut-off Amicon centrifugal concentrator to a final concentration of 20 mg/ml.

**Crystallization.** Purified 70.a proteins were subjected to hanging drop crystallization using commercial screens and Mosquito robotics (SPT Labtech). Crystals were directly harvested from 96-well plates and plunged briefly into 25% glycerol as a cryoprotectant. 70.a crystals grew from the commercial screen Morpheus containing 0.002 M Divalent II [0.005 M manganese(II) chloride tetrahydrate, 0.005 M cobalt(II) chloride hexahydrate, 0.005 M nickel(II) chloride hexahydrate, 0.005 M zinc acetate dihydrate], 0.1 M buffer system 6 pH 8.5 [Gly-Gly, AMPD], and 30% precipitant mix 7 [20% w/v PEG 8000, 40% v/v 1,5-pentanediol]. Crystals of purified 80.a and 80.b proteins were grown and harvested from commercial screens under numerous PEG/salt conditions. However, the best crystals were obtained with MES buffer at pH 6.5, with PEG3350 as the precipitant. See Supplementary Table S2 for Crystallographic Data and Refinement Statistics.

**Data collection and refinement.** The crystals were flash cooled in liquid nitrogen and subjected to X-ray data collection at the Advanced Photon Source (Argonne, Ill; NECAT beamline), or the NSLS2 synchrotron at Brookhaven, NY (FMX beamline) (Supplementary Table S1). Data were processed using XDS[68] and Aimless[69]. The structure of 70.a was determined by molecular replacement using WT $\beta$-lactamase as a search model in Phaser (PDB: 1XPB)[34,70]. The initial model was built using the program AutoBuild implemented in PHENIX[71]. This was followed by alternate cycles of manual model building in COOT[72] and refinement using PHENIX.

**Structural comparison to published $\beta$-lactamase homologs**
**Publicly-available $\beta$-lactamase structures.** Experimental $\beta$-lactamase structures were collected by performing a PSI-BLAST search (5 iterations) of the WT TEM-1 sequence with the database set to Protein Data Bank (PDB:). Structures annotated as beta lactamase were further filtered to remove those that had less than 20% sequence identity to WT TEM-1. After filtering, 927 polypeptide chains from 542 PDB structures remained.

**Structural alignment.** Chains were aligned in pymol using the "super" or "align" command, and the alignment with the lowest RMSD for each chain is reported and used to visualize structural variation.

**Software and code**
**Data collection.** The following software was used for data collection: Applied Biosystems Protein Thermal Shift software version 1.2, blastp 2.14.1, jackhmmer - 3.2.1, and MatLab 9.4 R2018a.

**Data analysis.** The following was used for data analysis: EVcouplings 0.1.2 (currently development version on github), Python 3.9.17, PyMOL 2.5.2, XDS February 5, 2021, Aimless 0.5.32, PHENIX 1.20.1, COOT 9.8.6, pandas 2.0.3, biopython 1.81, seaborn 0.12.2, openpyxI 3.1.2, scikit-learn 1.3.0, jupyter 1.0.0, Imfit 1.2.2, and numpy 1.23.5.

**Reporting summary**
Further information on research design is available in the Nature Portfolio Reporting Summary linked to this article.

## Data availability

There are no restrictions on data access. All source data used for figures in this study are provided in the Source Data file provided by the journal. All code, raw and processed data are available at https://github.com/gauthierscience/beta-lac-protein-design[73]. The crystal structures generated in this study have been deposited in the RCSB Protein Data Bank (RCSB PDB) under accession codes 8RQU [10.2210/pdb8RQU/pdb] (70.a), 8GII [10.2210/pdb8GII/pdb] (80.a), and 8GIJ [10.2210/pdb8GIJ/pdb] (80.b). The plasmids generated in this study have been deposited at Addgene under accession codes 202351 (WT TEM-1), 202350 (neg. ctrl), 202349 (rw-consensus), 202332 (98.a), 202333 (98.b), 202334 (95.a), 202335 (95.b), 202336 (90.a), 202337 (90.b), 202338 (80.a), 202339 (80.b), 202340 (70.a), 202341 (70.b), 202342 (50.a), 202343 (50.b), 202347 (opt.a), 202348 (opt.b). The nucleotide sequences used in this study are available in the Source Data and have been deposited at GenBank under accession codes PP763450 (WT TEM-1), PP763457 (neg. ctrl), PP763449 (rw-consensus), PP763456 (98.a), PP763452 (98.b), PP763460 (95.a), PP763447 (95.b), PP763455 (90.a), PP763453 (90.b), PP763461 (80.a), PP763448 (80.b), PP763458 (70.a), PP763446 (70.b), PP763445 (50.a), PP763459 (50.b), PP763451 (opt.a), PP763454 (opt.b). The WT TEM-1 crystal structure is available at RCSB Protein Data Bank (RCSB PDB) under accession code 1XPB [10.2210/pdb1XPB/pdb]. Accession codes for the natural multiple sequence alignment used for model generation as well as accession codes for the 542 PDB structures used in Fig. 6 are available in the Source Data. Source data are provided with this paper.

## Code availability

All data and custom analysis code are available at: https://github.com/gauthierscience/beta-lac-protein-design[73].

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

## Acknowledgements
We would like to thank Dan Davidi for advice relating to enzyme kinetics, Frank Poelwijk for insightful discussion, Lily Cerami for technical assistance, and Ethan Cerami for support. The Center for Macromolecular Interactions at Harvard Medical School for advice, discussion, and equipment. This work is based upon research conducted at the Northeastern Collaborative Access Team beamlines, which are funded by the National Institute of General Medical Sciences from the National Institutes of Health (P30 GM124165). This research used resources of the Advanced Photon Source, a U.S. Department of Energy (DOE) Office of Science User Facility operated for the DOE Office of Science by Argonne National Laboratory under Contract No. DE-AC02-06CH11357. This research also used the FMX beamline of the National Synchrotron Light Source II, a U.S. Department of Energy (DOE) Office of Science User Facility operated for the DOE Office of Science by Brookhaven National Laboratory under Contract No. DE-SC0012704. The Center for BioMolecular Structure (CBMS) is primarily supported by the National Institutes of Health, National Institute of General Medical Sciences (NIGMS) through a Center Core P30 Grant (P30GM133893), and by the DOE Office of Biological and Environmental Research (KP1607011). This material is based upon work supported by the U.S. Department of Energy, Office of Science, Office of Biological and Environmental Research, Genomic Science Program under Award Number DE-SC0022024 (N.P.G.), a grant from Science Foundation Ireland (SFI 20/FFP-A/8446) (A.R.K), SynBio HIVE at Harvard Medical School, and from Dana-Farber Cancer Institute (C.S. and N.P.G.).

## Author contributions
Project initiation and conceptualization: B.F., A.J.R., J.B.I., D.S.M., C.S., and N.P.G. Bench experiments and analysis: B.F., I.T., A.P., E.N., S.L., K.R., G.K., M.A.S., A.R.K., and N.P.G. Computational experiments and analysis: Y.S., A.J.R., J.B.I., and N.P.G. Project management: B.F. Supervision: D.S.M., A.R.K., C.D.B., C.S., and N.P.G. Manuscript writing: B.F., Y.S., J.B.I., N.N.T., A.R.K., C.S, and N.P.G. All authors reviewed and approved the manuscript.

## Competing interests
The authors declare no competing interests.
