## [Peer Review File · Nature Communications]

Reviewers' Comments:

Reviewer #1:

Remarks to the Author:

The study by Fram et al. addresses the computational design of proteins with enhanced functional properties through the use of evolutionary models. They computationally design diverse variants of the TEM-1 β -lactamase protein using co-evolutionary analysis through the EVCouplings model.

Experimental testing of 14 sequences revealed that 12 out of 14 designed variants, including one with 84 mutations, were functional and exhibited increased thermostability and promiscuous activity in degrading multiple antibiotics. This work demonstrates the effectiveness of evolutionary models in guiding large sequence changes while maintaining functional diversity in protein design. The work is carefully designed, well-written, with a lucid presentation of its findings. However, it could be enhanced by addressing the issues discussed below:

1. It is interesting that the majority of the selected mutations by the Gibbs Sampling based on EVH are usually located on the surface of the proteins. Given that the surface proteins are less conserved, this may be due to evolutionary rates inferred from the MSA by the Pott Model (the core of the EVCouplings). Thus, it would be interesting to explore the correlation between mutational rates per position and their conservation or evolutionary rates. They showed this qualitatively in Figure 2C, but a quantitative analysis could be useful.

2. The authors compared the predicted all single-point mutations using EVH scores with the experimental deep sequencing data by Ranganathan group. However, the wild type (WT) TEM1 sequence in this work and the WT sequence in Ranganathan's work (1btl.pdb) are slightly different with a few substitutions. It is not clear if they used the correct sequence; if not, it is better to use the correct sequence.

3. Likewise, the detected non-neutral substitution could be different in those two background sequences, as discussed above. This should be checked. It would be interesting to work on using EVH scores to see which set of mutations compensate for the deleterious effect of the non-neutral substitutions.

4. The authors should also provide pairwise alignment of the experimentally characterized sequences with the close homologs. Furthermore, it would be interesting to investigate if the designed sequences exhibit similar biophysical properties to the close homologs. Thus, the authors should report biophysical properties (i.e., thermal stability, MIC values) of the close homologs.

5. The details of the method are missing. How fast is it? How many iterations are necessary to reach the targeted proteins starting from a random sequence? What is the temperature used for MC? The optimum sequences are obtained using parallel tempering; again, details are missing. Also, the available code on GitHub fails when I try to run it. A Read me file should be added.

6. While the work clearly shows successful design of enzymes, similar approaches have been used by Morcos and Best groups. Robert Best and his co-workers used a similar approach to design three small proteins (see <https://doi.org/10.1002/ange.201713220>) and showed that they fold and were able to bind to native ligands, in some cases with higher affinity than wild-type.

Interestingly, they later applied it to design switch proteins (<https://doi.org/10.1371/journal.pcbi.1008285>). Similarly, Morcos used a similar approach to design RNA sequences (doi: 10.1038/s41467-018-04729-0) to improve binding affinity and to design novel hybrid repressor proteins to modulate expression (doi: 10.1038/s41467-021-25851-6). These papers should be cited, and differences in the approaches should be discussed.

7. The abstract states that "Nearly all designed variants were functional." However, only 14 sequences are experimentally tested. So it should be replaced with "Nearly all characterized design sequences are functional."

8. As minor points, in the method section explaining obtaining Ampicillin kinetic parameters, it is written as "each design-nitrocefin concentration a linear regression." Nitrocefin should be replaced by ampicillin.

Reviewer #2:

Remarks to the Author:

The manuscript entitled "Simultaneous Enhancement of Multiple Functional Properties Using Evolution-Informed Protein Design" by Fram, et al. introduces an interesting approach to designing functional proteins, using the enzyme beta-lactamase as a test case. This approach utilizes a MSA

generated for a starting protein of interest with the desired function, and uses a relatively straightforward sampling algorithm that accounts for sequence conservation, residue co-variation, and a maximum sequence identity threshold to generate highly divergent variants that retain the function of the input protein. The authors apply this strategy to the TEM-1 beta-lactamase, and generate 42 sequences with varying sequence identity relative to TEM-1 (50-96.2%), which are filtered down to 14 sequences that were rigorously tested experimentally for beta-lactamase activity in vivo and in vitro, as well as analyzed structurally. Remarkably, 11 of the 14 sequences tested show beta-lactamase activity, the majority also have increased thermostability and broadened substrate specificity, and crystal structures of three designs show a high degree of structural similarity to TEM-1. Some of these mutants have as many as 84 mutations relative to the parent TEM-1 sequence. In contrast, other studies have shown a sharp decrease in protein function with the accumulation of a small number of mutations. As an important control, the authors also assess the consensus sequence from the MSA used as input for their design algorithm. This consensus sequence has increased thermostability, but poor function in beta-lactamase assays, which demonstrates the importance of the sequence co-variation information in preserving function despite the introduction of many individual mutations. The authors do an excellent job of demonstrating the complex epistasis that their method exploits by observing how specific mutations compensate for the deleterious effects of the G251W in TEM-1, a mutation that was present in two of their designs, and also by performing some clever calculations to show that the predicted fitness effects of individual mutations are context dependent (i.e. mutations that increase fitness in the background of the designs typically decrease fitness in the wild type TEM-1 background).

Overall, I think this is an excellent manuscript that warrants publication with only minor revisions needed. The manuscript is well written, and presents a protein design algorithm that is fundamentally interesting and appears to work incredibly well, even if limited in scope (the method appears to only be useful for designing proteins with functions that already occur in nature). The designed enzymes reported here are rigorously validated using multiple experimental techniques, which is a major strength of the work. Furthermore, I think the paper will be of interest to a broad readership, spanning the fields of protein design, genetics, structural biology, and enzymology. I have the following minor comments, which the authors should address in the final version:

- My biggest concern has to do with technical aspects of the X-ray crystallography. First, the authors state that the structures were determined using molecular replacement with WT beta-lactamase as a search model. They determine structures that have very low coordinate RMSD to TEM-1, but it is not stated whether they have done anything to attempt to remove model bias from the electron density maps that were used to build the structures. It seems sensible to perform some type of coordinate perturbation (shaking, simulated annealing, etc.) prior to initial refinement in attempt to remove any possible model bias. It is not stated whether this has been done. I think this is especially relevant in the context of the 70.a structure, which is determined at low resolution (3.11 Å) and has a large Rfree-Rwork gap of 0.047. Second, it appears that very conservative resolution cutoffs have been taken for the 70.a and 80.b structures, with $I/\sigma(I)$ exceeding 3.0 and CC1/2 exceeding 0.8 in the high-resolution shells. This contrasts with the treatment of the 80.a data, which appears to conform to more modern standards of resolution cutoffs, especially given that the data were collected at synchrotron beamlines with sensitive pixel array detectors. Third, the authors do not list the alpha, beta, and gamma unit cell angles in supplemental table 1. These are trivial (90 degrees) for the tetragonal and orthorhombic space groups, but not for the triclinic space group, so they should be included, at least for the 80.a crystal (space group P1).
- The structural similarity of the designs to TEM-1 are assessed using a global RMSD metric, which can be a poor metric of structural variation, because large, local changes can be obscured by regions that are well-conserved structurally. I suggest the authors choose a method of comparison that can better capture local variation, such as a difference distance matrix. Additionally, I wonder whether there are other analyses that could be performed on the crystal structures to gain additional insight, especially with respect to the promiscuity of the designs. For example, do the designs have increased active site SASA or B-factors?
- The authors mention in the discussion the relationship of this work to other ML-based

approaches. I think it could be worth mentioning the work by Madani, et al. (<https://pubmed.ncbi.nlm.nih.gov/36702895/>) in which a LLM is used to generate highly divergent lysozyme sequences that have activities similar to natural sequences. This seems like a direct parallel to the work presented. Additionally, work by Goldenzweig, et al. (<https://www.ncbi.nlm.nih.gov/pmc/articles/PMC4961223/>) demonstrates a method for dramatically enhancing thermostability while retaining/enhancing functional properties, which operates using a MSA and generates divergent sequences with dozens of mutations relative to the input sequence. Both studies seem highly relevant to the work presented in this manuscript, and their mention could add additional insight to the Discussion section.

- I am curious whether the authors think that adding structural information into the design algorithm could be advantageous, such as when sequences in the MSA also have known structures. This could be another idea to mention in the discussion.
- The controls used for the design analysis are the wild type TEM-1 and a consensus sequence derived from the MSA. Were any of the other naturally occurring beta-lactamases represented in the alignment assayed? This might make another interesting comparison. I also wonder about the consensus of the designs in comparison to the consensus from the MSA. Additional experiments are certainly not required for the manuscript to be published, but these would be interesting data to add if the authors choose.
- Finally, I note that some of the figures contain a decent amount of small text. Please confirm with editorial staff that these will appear large enough to be easily readable.

Reviewer #3:

Remarks to the Author:

In this report, Fram et al. describe the use of evolutionary models of sequence co-variation (EV couplings) to computationally design TEM1 beta lactamase variants with high sequence divergence from natural lactamases. In an initial step they generated a maximum entropy model, which includes site specific and pairwise constraints based on a multiple sequence alignment of the protein of interest. Using a sampling algorithm, which optimises fitness of the entire sequence, they could also define the sequence distance to the protein of interest as well as the nearest homologues. For testing their sequence design algorithms, they chose TEM1 beta lactamase, which is an ideal model system - as it is an intensely studied and well characterised model system. The authors generated several sequences for multiple sequence identity thresholds (98%, 95%, 90%, 80%, 70%, 50%). In addition, they also designed two additional mutants that were not constrained by sequence identity. Two designs for each sequence threshold, as well as the additional designs were experimentally tested and characterised. Out of the 14 designed sequences, 11 sequences conferred ampicillin resistance to *E. coli*, with 8 designs showing similar resistance as WT TEM-1 (70b, 50.a, 50.b being inactive). Additional growth assays showed that some of the designs are active against antibiotics, towards which WT-TEM1 is inactive. The authors performed steady state kinetic analysis of the purified enzymes with nitrocefin as a substrate. Most designs showed either improvements in k_{cat} or K_M . Overall, multiple designs showed catalytic efficiencies that are higher than wild type TEM1. Using purified enzymes the authors also confirmed that the majority of their designs were capable of hydrolysing ampicillin with specific activities above WT-TEM1. Furthermore, the melting temperatures of their designs were increased by 10-20 °C compared to WT-TEM1. Structural analysis of 80.a, 80.b and 70.a by X-ray crystallography revealed that these proteins are in good agreement with WT-TEM1 and that structural deviations are in agreement with those of related lactamases with a similar level of sequence identity. Finally, the authors analysed the fitness contributions of individual mutations and showed that some of the introduced mutations are in fact on their own deleterious in WT TEM1 background, whereas in context of the design sequence they contribute to increased fitness. Overall, this is a very interesting and very well executed piece of work. In silico sequence design and diversification of protein sequences is an important challenge in protein design. Various algorithms and design approaches with a similar goal have been implemented over time, including PROSS, Fireprot, ProteinMPNN or evolution based models (Ref.2), which are based on sequence alignments or structure prediction. The method outlined in this manuscript is a valuable addition as

the results presented in this work are impressive - up to 88 mutations were inserted into a natural protein, increasing protein stability, while maintaining enzyme activity. The manuscript is well written and the figures are well designed and informative.

I believe this work will be of interest to the broad readership of Nature Communications and I strongly recommend publication.

A few minor points need to be addressed:

1) In vitro ampicillin hydrolysis by opt.b was not determined as the authors claimed, this is due to opt.b being prone to precipitation. However Michaelis Menten parameters for nitrocefin could be measured for opt.b, suggesting that opt.b can be characterised in vitro. It would be thus desirable to determine the missing ampicillin hydrolysis value.

2) What were the protein expression levels (or yields)? These values are not mentioned in the manuscript. In addition, it is mentioned that 50.a and 50.b could not be purified - does this mean they could not be expressed?

3) Melt curves and Michaelis Menten plots are missing in the SI.

4) Regarding the in vivo resistance assays, would it be possible to perform measurements at higher antibiotic concentrations, as WT-TEM1 and several of the designs show resistance at the highest chosen ampicillin concentration? The designs might compare even more favourable to the WT-TEM1.

5) It is unclear what the authors mean by "saturated concentration of ampicillin". 800 μ M does not seem saturated. Also, in the method section "Ampicillin" there seems to be a mistake, as parts of the nitrocefin section are repeated.

6) Figure 3C: The units on the y-axis are missing.

7) In the last part of the paper, the authors computationally analyse the contribution to fitness of the designed mutations and compare them to the results of a deep scanning experiment. Additional experimental data (characterisation of a few point mutants) could further confirm predictions/conclusions and validate their approach, making this an even stronger manuscript.

8) I encourage the authors to cite a recent ProteinMPNN preprint from the Baker lab in their manuscript: <https://doi.org/10.1101/2023.10.03.560713>

9) The criteria for choosing the designs for characterisation should be mentioned in the main text - not the methods (Sequence subsampling). In this regard, was the 90% criteria chosen, to ensure that the catalytic residues are maintained?

10) With regard to point 9, the question also raises, how efficient is the use of EV-couplings in maintaining the catalytic residues? It was surprising that the neg. ctrl. S70A did not have a devastating effect on predicted fitness. What about other active site residues, were these conserved in the designs? It would be worthwhile to also indicate active site positions in figure 2C or 2B (e.g. sequence logo). Overall, how well does the presented method conserve active site residues and therefore activity - or is calculated fitness primarily based on structure/stability.

11) Every sequence contained the M182T mutation, which contributes to stability in WT-TEM1. Was this coincidence? Did all of the six designs of each threshold have this mutation? Or was there potential bias in the MSA. What is the fitness prediction of this mutation (SI Fig. 1 suggest that there only very few single mutants with increased predicted fitness)? Does EV-coupling primarily report on structure rather than activity?

12) k_{cat} , K_M and T_m are incorrectly formatted throughout the manuscript.

13) In the analysis of the X-ray structures towards rationalising T_m increases, the authors tallied

the number of hydrogen bonds and atom pair contacts but could not find significant differences. Did the authors analyse the B-factors of the structures?

14) Introduction: Directed evolution is not a "common approach to protein design" - rather the contrary. It might also be worthwhile to include that directed evolution is costly and time consuming.

15) Figure 2: A: there are gaps on the blue bars at around position 58 and 239 - is this intentional? Furthermore, the description of the legend is unclear. Are the blue dots PDB contacts that coincide with model couplings? B: How are core residues defined? C: There are gaps in the sequence logo and the conservation plot - was this intentional? What is the meaning of lower-case letters in the WT sequence?

16) The kinetic parameters for nitrocefin are likely below those of ampicillin for WT-TEM1, as nitrocefin is not the actual substrate. Is there any data in the literature that compares/correlates the ampicillin to nitrocefin hydrolysis in WT-TEM1?

17) SI Figure 8: It is unclear to what "functional design" refers to in the legend?

18) It would be desirable, if the authors could discuss in the manuscript, why consensus design and EV-couplings increases protein stability.

19) Additional images of structural alignments in the SI would be desirable, to highlight differences to WT-TEM1 (as figure 6C is hard to read), in particular of the active site to highlight the mutations that enable activity towards aztreonam.

20) The summaries after each paragraph are unnecessary and should be moved to the conclusion section.

21) Sequences of the construct (e.g. DNA sequences) are not included in the SI.

Reviewer #4:

Remarks to the Author:

Fram et al. present a highly interesting study wherein evolutionary information was used to design variants of TEM1 beta lactamase with enhanced stability and broadened substrate specificity. They computationally constructed and experimentally characterized fourteen variants of TEM-1 B-lactamase with varying sequence identity to the native TEM-1, and showed that the vast majority of these variants maintained activity while increasing stability. The work is an important addition to the somewhat limited literature that experimentally tests the capacity of evolution-based design algorithms. The experiments are well done, and it is nice to see the accompanying structures. Critically their results show both where the method succeeds and breaks down. The results suggest high potential for a "jump and walk" laboratory evolution strategy, wherein protein designs are used as new starting points for the evolution of enhanced or novel function. The manuscript is nicely written, the data analysis is rigorously presented, and the accompanying github repository appears reasonably well-organized. We anticipate that this work will be of broad interest to the protein engineering and design community, some enzyme biochemists, and those specifically working on the TEM-1 model system. While we are enthusiastic about the work as a whole, we have two major concerns and a few minor concerns.

Major concerns:

1)The authors state that "surprisingly, all functional designs had large increases in thermostability", which seems to overlook prior work suggesting that evolutionary models (particularly Potts-like models, including DCA and EVcouplings) show some correlation between favorable score and increased thermodynamic stability. For example, see: Morcos et al (2014) "Coevolutionary information, protein folding landscapes, and the thermodynamics of natural selection" PNAS 111:12408. And even more critically: Figliuzzi et al (2015) "Coevolutionary

landscape inference and the context dependence of mutations in beta-lactamase TEM-1" Mol Biol Evol 33:268. Figliuzzi et al. constructs a highly similar Potts-style model of the TEM-1 beta-lactamase family, and in figure 4 shows some correlation between model score and thermal stability. These prior results support and indeed almost predict the authors' results that evolutionary models resulted in more thermal stable designed proteins. It would improve the manuscript to more thoroughly discuss the connection between this earlier work and the authors' own findings. It would likewise be interesting to hear any ideas from the authors about why EVcouplings designs are more stable. The authors found that the increased thermal stability is not explained by increased hydrogen bonding, and they found that all designs contained known stability-enhancing mutants. Additional questions of interest could include: Does the TEM-1 alignment include a large fraction of sequences from thermophiles? Would removing thermophilic sequences from the alignment result in fewer stability-enhanced designs? Do the designs select mutations that improve the site or coupling parameters portion of the model score? Further commentary on how the EVcouplings model enhances stability may provide the authors an opportunity to contribute to the body of prior work in this area.

2) One aspect of evolution-informed computational protein design is the choice of algorithm for inferring site and coupling parameters given a sequence alignment. EVcouplings uses regularized maximum pseudolikelihood, however other strategies exist (mean field, ACE, Boltzmann Machine). These strategies vary in the fidelity with which they recapitulate pairwise (and higher order) statistics of the natural alignment (e.g. Fig 2 of Cocco et al. (2018) "Inverse statistical physics of protein sequences: a key issues review" Reports on Progress in Physics). How well does the inferred EVcouplings model recapitulate the pairwise alignment statistics of the natural alignment (i.e. if you compare between an alignment of designed and natural sequences)? It would be nice to have a plot of this alongside the other model validation strategies (showing recovery of contacts and prediction of mutant effects). Additionally, could the authors spend some time to discuss what might be improved to push designs beyond the apparent 70% identity threshold? Is this limitation coming from the inference strategy, sampling size limitations, a need for higher order statistics, or perhaps something else entirely?

Minor concerns:

- 1) It would be useful to indicate the number of designed sequences that were experimentally characterized in the abstract
- 2) Again in the abstract: this is probably more a matter of taste, but I would not typically consider a sequence with 70% identity to the WT as "highly divergent". Maybe "highly mutated" or "with many mutations"?
- 3) On page 4, a sentence or two briefly explaining greedy sampling and parallel tempering would help a broader audience with less expertise in optimization.
- 4) On page 10, 9 lines from bottom, "an 7.5C increase" should be changed to "a 7.5C increase"
- 5) Please provide an equation for the Gibbs sampling cost function in the methods section
- 6) Supplemental figure 3b legend states that "Residues mutated in the designs were less likely to be surface accessible than non-mutated residues". This seems to contradict the figure and what is written in the main text ("the algorithm preferred mutating positions that were closer to the surface")
- 7) It would be helpful to indicate the active site in Figure S4

Reviewer #5:

Remarks to the Author:

Simultaneous Enhancement of Multiple Functional Properties Using Evolution-informed Protein Design

Point-by-point responses to reviewers' comments:

We thank all reviewers for their constructive and insightful critiques. We believe that the paper has been significantly strengthened from all of these reviews.

Reviewer #1: The study by Fram et al. addresses the computational design of proteins with enhanced functional properties through the use of evolutionary models. They computationally design diverse variants of the TEM-1 β -lactamase protein using co-evolutionary analysis through the EVcouplings model. Experimental testing of 14 sequences revealed that 12 out of 14 designed variants, including one with 84 mutations, were functional and exhibited increased thermostability and promiscuous activity in degrading multiple antibiotics. This work demonstrates the effectiveness of evolutionary models in guiding large sequence changes while maintaining functional diversity in protein design. The work is carefully designed, well-written, with a lucid presentation of its findings. However, it could be enhanced by addressing the issues discussed below:

1. It is interesting that the majority of the selected mutations by the Gibbs Sampling based on EVH are usually located on the surface of the proteins. Given that the surface proteins are less conserved, this may be due to evolutionary rates inferred from the MSA by the Pott Model (the core of the EVCouplings). Thus, it would be interesting to explore the correlation between mutational rates per position and their conservation or evolutionary rates. They showed this qualitatively in Figure 2C, but a quantitative analysis could be useful.

We did observe a tendency to mutate surface rather than core residues, although many of the mutations are in the core of the protein. As with Figure 2C, these data are in part visualized in the surface accessibility CDF plot of Supplemental Figure S4B, which we have now updated to include all generated designs (not just those that were tested). From your suggestion, we added a new plot in the manuscript (Supplemental Figure S5) that, for each position, compares mutation count to conservation and surface accessibility, which provides some insight into the selection of mutations.

2. The authors compared the predicted all single-point mutations using EVH scores with the experimental deep sequencing data by Ranganathan group. However, the wild type (WT) TEM1 sequence in this work and the WT sequence in Ranganathan's work (1btl.pdb) are slightly different with a few substitutions. It is not clear if they used the correct sequence; if not, it is better to use the correct sequence.

The wild type sequences are identical between WT TEM1 from the paper and the deep mutational scan from see Stiffler et al. 2015 (see Table S1). However (1) they have a different numbering scheme from the model and (2) 11 positions are poorly aligned in the multiple sequence alignment (six N-terminal amino acids - HPETLV, three C-terminal amino acids -

KHW, and two additional amino acids E58 and R241), and are therefore ignored by the model and not mutated in the designs. However, Stiffler et al. did test mutations at these positions. We removed these unaligned positions from Supp Fig 8 when plotting, and have now noted as such in the figure legend. Although the WT TEM1 sequence used in this work and the sequence tested by Stiffler et al., 2015 are identical, there are indeed two two substitutions (V84I and A184V) relative to these sequences in the referenced 1bt1 pdb file.

3. Likewise, the detected non-neutral substitution could be different in those two background sequences, as discussed above. This should be checked. It would be interesting to work on using EVH scores to see which set of mutations compensate for the deleterious effect of the non-neutral substitutions.

The WT TEM-1 sequence and the sequence tested by Stiffler et al., 2015 are identical (see response to #2 above).

4. The authors should also provide pairwise alignment of the experimentally characterized sequences with the close homologs. Furthermore, it would be interesting to investigate if the designed sequences exhibit similar biophysical properties to the close homologs. Thus, the authors should report biophysical properties (i.e., thermal stability, MIC values) of the close homologs.

The idea that the designs “borrowed” properties from their closest homologs in the multiple sequence alignment is very interesting. The objective function used for Gibbs Sampling optimization does enforce a sequence identity upper bounds relative to any homolog in the multiple sequence alignment, which results in WT TEM-1 usually being the closest homolog in the MSA. However, because the constraint is implemented as a range (e.g., 70% aims for 65-70% sequence identity to WT TEM-1 and maximum of 70% identity to any other sequence in the MSA), it is possible for the homologs to have fewer mutations. In our dataset, this only occurs in one of the tested distance-constrained designs (i.e., the non-functional design 70.b contains 88 mutations from WT TEM-1 and there are two sequences in the MSA with fewer mutations from 70.b: one with 87 mutations and the other with 84). Although experimental characterization of these homologs is interesting, we feel it is beyond the scope of this manuscript. However, per the suggestion, we have now added a new figure (Supplemental Fig S3) that contains pairwise alignments of each distance-constrained design together with the top five most similar sequences from the MSA used for model inference.

5. The details of the method are missing. How fast is it? How many iterations are necessary to reach the targeted proteins starting from a random sequence? What is the temperature used for MC? The optimum sequences are obtained using parallel tempering; again, details are missing. Also, the available code on GitHub fails when I try to run it. A Read me file should be added.

Thank you for pointing out these omissions. We have updated the Github code and re-tested its ability to run on new installations together with a README file. This code was also tested and

successfully executes on MATLAB online. In addition, we have added more detail including speed, iterations, temperature, and details about parallel tempering.

6. While the work clearly shows successful design of enzymes, similar approaches have been used by Morcos and Best groups. Robert Best and his co-workers used a similar approach to design three small proteins (see <https://doi.org/10.1002/ange.201713220>) and showed that they fold and were able to bind to native ligands, in some cases with higher affinity than wild-type. Interestingly, they later applied it to design switch proteins (<https://doi.org/10.1371/journal.pcbi.1008285>). Similarly, Morcos used a similar approach to design RNA sequences (doi: 10.1038/s41467-018-04729-0) to improve binding affinity and to design novel hybrid repressor proteins to modulate expression (doi: 10.1038/s41467-021-25851-6). These papers should be cited, and differences in the approaches should be discussed.

These are very relevant and interesting papers, and we have added references and discussed them in the main text.

7. The abstract states that "Nearly all designed variants were functional." However, only 14 sequences are experimentally tested. So it should be replaced with "Nearly all characterized design sequences are functional."

We have updated the abstract with the more precise language.

8. As minor points, in the method section explaining obtaining Ampicillin kinetic parameters, it is written as "each design-nitrocefin concentration a linear regression." Nitrocefin should be replaced by ampicillin.

Thank you for discovering this typo. We have updated the main text.

Reviewer #2: The manuscript entitled "Simultaneous Enhancement of Multiple Functional Properties Using Evolution-Informed Protein Design" by Fram, et al. introduces an interesting approach to designing functional proteins, using the enzyme beta-lactamase as a test case. This approach utilizes a MSA generated for a starting protein of interest with the desired function, and uses a relatively straightforward sampling algorithm that accounts for sequence conservation, residue co-variation, and a maximum sequence identity threshold to generate highly divergent variants that retain the function of the input protein. The authors apply this strategy to the TEM-1 beta-lactamase, and generate 42 sequences with varying sequence identity relative to TEM-1 (50-96.2%), which are filtered down to 14 sequences that were rigorously tested experimentally for beta-lactamase activity in vivo and in vitro, as well as analyzed structurally. Remarkably, 11 of the 14 sequences tested show beta-lactamase activity, the majority also have increased thermostability and broadened substrate specificity, and crystal structures of three designs show a high degree of structural similarity to TEM-1. Some of these mutants have as many as 84 mutations relative to the parent TEM-1 sequence. In contrast,

other studies have shown a sharp decrease in protein function with the accumulation of a small number of mutations. As an important control, the authors also assess the consensus sequence from the MSA used as input for their design algorithm. This consensus sequence has increased thermostability, but poor function in beta-lactamase assays, which demonstrates the importance of the sequence co-variation information in preserving function despite the introduction of many individual mutations. The authors do an excellent job of demonstrating the complex epistasis that their method exploits by observing how specific mutations compensate for the deleterious effects of the G251W in TEM-1, a mutation that was present in two of their designs, and also by performing some clever calculations to show that the predicted fitness effects of individual mutations are context dependent (i.e. mutations that increase fitness in the background of the designs typically decrease fitness in the wild type TEM-1 background).

Overall, I think this is an excellent manuscript that warrants publication with only minor revisions needed. The manuscript is well written, and presents a protein design algorithm that is fundamentally interesting and appears to work incredibly well, even if limited in scope (the method appears to only be useful for designing proteins with functions that already occur in nature). The designed enzymes reported here are rigorously validated using multiple experimental techniques, which is a major strength of the work. Furthermore, I think the paper will be of interest to a broad readership, spanning the fields of protein design, genetics, structural biology, and enzymology. I have the following minor comments, which the authors should address in the final version:

1. My biggest concern has to do with technical aspects of the X-ray crystallography. First, the authors state that the structures were determined using molecular replacement with WT beta-lactamase as a search model. They determine structures that have very low coordinate RMSD to TEM-1, but it is not stated whether they have done anything to attempt to remove model bias from the electron density maps that were used to build the structures. It seems sensible to perform some type of coordinate perturbation (shaking, simulated annealing, etc.) prior to initial refinement in attempt to remove any possible model bias. It is not stated whether this has been done. I think this is especially relevant in the context of the 70.a structure, which is determined at low resolution (3.11 Å) and has a large Rfree-Rwork gap of 0.047. Second, it appears that very conservative resolution cutoffs have been taken for the 70.a and 80.b structures, with $1/\sigma(I)$ exceeding 3.0 and CC1/2 exceeding 0.8 in the high-resolution shells. This contrasts with the treatment of the 80.a data, which appears to conform to more modern standards of resolution cutoffs, especially given that the data were collected at synchrotron beamlines with sensitive pixel array detectors. Third, the authors do not list the alpha, beta, and gamma unit cell angles in supplemental table 1. These are trivial (90 degrees) for the tetragonal and orthorhombic space groups, but not for the triclinic space group, so they should be included, at least for the 80.a crystal (space group P1).

Thank you for the thoughtful and detailed comment.

- Following MR, the first step was Autobuild implemented in Phenix. Cartesian simulated annealing was used during this initial model building step to reduce model bias. More

detailed information about the refinement are shown in Methods and Table 1, including the unit cell angles.

- During all stages of model building, the electron density maps were inspected to verify the sequence changes in b-lactamase.
- Following this suggestion, data for 70.a was extended to 2.90 Angstrom resolution. There is a rapid dropoff in intensities, but we found these data useful. MR and refinement were performed again from the beginning, to reduce model bias with the higher resolution data. We were able to build a glycine-rich b-strand that was missing in the original 3.1 Angstrom model (residues 54-59), and a loop 225-229. We appreciate the comments of the reviewer, since the model 70.a is now more complete.
- Regarding the 80.b data set, unfortunately we could not recover the data that presumably extends beyond 1.59 Angstrom as the crystal-to-detector distance was set for 1.6 Ang data collection (maximum resolution).
- Finally, we have calculated composite omit maps of the three refined structures (Supplemental Figure S19). These omit maps provide a means of assessing model bias in refined structures. Cartesian simulated annealing was performed together with systematic deletions in 70.a, as implemented in Phenix. The electron density of 70.a around Trp250 is shown as an example. For 80.a/80.b, the starting MR model is fully superimposed using secondary structure matching in Coot. Sections of the omit map confirm that electron density corresponds to the refined models, and not the starting model 3cmz.

2. The structural similarity of the designs to TEM-1 are assessed using a global RMSD metric, which can be a poor metric of structural variation, because large, local changes can be obscured by regions that are well-conserved structurally. I suggest the authors choose a method of comparison that can better capture local variation, such as a difference distance matrix. Additionally, I wonder whether there are other analyses that could be performed on the crystal structures to gain additional insight, especially with respect to the promiscuity of the designs. For example, do the designs have increased active site SASA or B-factors?

Thank you for these suggestions.

Difference Distance Matrices: We have now generated difference distance matrices for each of the structurally-determined designs (70.a, 80.a, 80.b) and have added these figures to the paper (Supplemental Figures S13, S14, S15). In short, these matrices revealed few distance differences between residue pairs, with the exception of two looped regions (positions ~53-55 and ~255-257). These analyses support the idea that the designs are both globally and locally similar to WT TEM-1.

We also performed analyses of SASA and B-factors with an aim at trying to describe mechanisms for increased promiscuity and thermostability. There are three PDB entries (1XPB, 4GKU, and two chains of 1S0W) with exactly the same amino acid sequence as the WT TEM-1 used in this work, and these were used as controls for normal variation. A detailed description is below, but in short these analyses do not provide obvious insight into the mechanism for these

enhanced properties. We have also included a short description of these analyses in the main text.

SASA analysis: The Relative SASA of all active site pocket residues (withing 4Å of S70 or E166) are all in agreement with at least one of the four WT TEM-1 PDB structures:

Active Site Pocket	Relative SASA (% of maximal SASA for each residue)																			
	68	69	70*	71	72	73	103	130	132	136	165	166*	167	168	169	170	234	235	236	
WT TEM-1	M	M	S	T	F	K	V	S	N	N	W	E	P	E	L	N	K	S	G	
1xpb_A	1	0	6	0	0	0	25	7	6	12	34	1	18	44	1	10	0	1	2	
4gku_A	1	0	8	0	0	0	23	8	8	11	35	1	36	52	1	10	0	3	3	
1s0w_A	1	0	8	0	0	0	22	10	7	10	36	1	21	53	2	13	0	3	3	
1s0w_B	1	1	8	0	0	0	25	9	6	10	36	1	21	50	1	13	0	4	4	
70.a_A	1	1	8	0	0	1	22	6	7	13	36	4	37	41	1	11	0	2	1	
70.a_B	1	2	9	0	0	0	22	6	8	11	36	3	44	36	2	15	0	2	1	
80.a_A	1	0	6	0	0	0	28	8	6	12	35	2	29	54	1	10	0	3	2	
80.a_B	0	1	6	0	0	0	27	8	7	11	34	1	31	55	1	10	0	2	2	
80.b_A	0	0	6	0	0	0	27	9	7	11	41	4	34	45	1	13	0	3	3	
80.b_B	0	0	8	0	0	0	24	9	8	9	40	3	35	45	1	12	0	4	4	

We also attempted to analyze individual residue SASA differences (both relative and absolute) between WT structures and designs, with a focus on the active site (see **Reviewer Response Addendum Figure 1 below**). Although there were differences for individual residues, none of these differences were consistent (i.e., had the same difference between all WT structures and the design). This may in part be because residues in the wild type structures also have a great deal of variation.

B-factor analysis: It is difficult to compare B-factors between structures because of resolution differences, and the resolution of our structures and the four published WT TEM-1 structures is different. The mean B is approximately equivalent to the Wilson, which describes how the intensity of the data falls off as a function of resolution. Further compounding analysis is the fact that B-factors are on individual atoms (not residues) and so comparisons of mutation versus wild type residue require analysis of either a shared atom (e.g., C α atom) or a function of the set of sidechain atoms (e.g., mean). As expected due to resolution differences, we see a large variance of absolute B-factor statistics for the design and wild type structures:

	B factors, CA				B factors, sidechain			
	min	max	mean	std	min	max	mean	std
1xpbA	7.01	27.21	14.57	4.052	6.403	41.09	17.93	7.242
4gkuA	7.18	30.71	14.63	4.595	7.74	39.89	17.02	7
1s0wA	13.59	45.38	27.57	7.519	13.13	56.05	28.9	9.358
1s0wB	13.49	56.5	29.9	10.09	10.33	60.58	30.9	11.39
70aA	30.78	94.09	48.718	11.436	28.71	94.71	49.15	11.976
70aB	45.13	120.52	75.907	19.374	45.327	125.42	75.786	18.964
80aA	8.89	50.02	18.04	6.075	9.13	108.2	20.76	9.682
80aB	10.07	57.19	18.72	6.717	8.58	153.6	21.8	12.13
80bA	7.23	26.23	12.92	3.714	8.01	77.33	16.61	7.816
80bB	7.74	53.12	12.8	4.373	9.303	51.11	15.74	6.27

We also attempted to analyze individual residue B-value differences between WT structures and designs (both C α atom and average of all side chain atoms), with a focus on the active site (see **Reviewer Response Addendum Figure 2 below**). Although there were differences, as with the SASA analysis, none of these differences were consistent (i.e., had the same difference

between all WT structures and the design). Again similar to the SASA analysis, the lack of consistent differences may in part be because the wild type vs wild type structures also have a great deal of variation.

3. The authors mention in the discussion the relationship of this work to other ML-based approaches. I think it could be worth mentioning the work by Madani, et al. (<https://pubmed.ncbi.nlm.nih.gov/36702895/>) in which a LLM is used to generate highly divergent lysozyme sequences that have activities similar to natural sequences. This seems like a direct parallel to the work presented. Additionally, work by Goldenzweig, et al. (<https://www.ncbi.nlm.nih.gov/pmc/articles/PMC4961223/>) demonstrates a method for dramatically enhancing thermostability while retaining/enhancing functional properties, which operates using a MSA and generates divergent sequences with dozens of mutations relative to the input sequence. Both studies seem highly relevant to the work presented in this manuscript, and their mention could add additional insight to the Discussion section.

Thanks for the pointer to these interesting papers. We agree they are very relevant and have mentioned them in the discussion.

4. I am curious whether the authors think that adding structural information into the design algorithm could be advantageous, such as when sequences in the MSA also have known structures. This could be another idea to mention in the discussion.

EVcouplings and similar models do in part capture structural information as evidenced by their ability to identify interactions, ultimately providing a means to determine 3d structure. However, these statistical interactions don't comprehensively capture structure, and the addition of structural information may provide a means to further enhance the design methodology providing either higher yields or more performant designs. There are several ways structural information could be included including prioritization in the objective function or filtering out structurally-unlikely candidate designs. We have added a sentence in the discussion about this idea.

5. The controls used for the design analysis are the wild type TEM-1 and a consensus sequence derived from the MSA. Were any of the other naturally occurring beta-lactamases represented in the alignment assayed? This might make another interesting comparison. I also wonder about the consensus of the designs in comparison to the consensus from the MSA. Additional experiments are certainly not required for the manuscript to be published, but these would be interesting data to add if the authors choose.

The complete set of sequences we characterized is presented, i.e., we did not test any additional beta-lactamases, or the consensus of the designs. We believe that the consensus of the functional designs is more likely than the consensus of all the designs (including those non-functional) to retain or enhance function, but it is beyond the scope for this work to perform such a detailed comparison. Future work will be aimed at asking whether the strategy presented here (or other design strategies) are more efficient at finding enhanced function relative to

randomly selected homologs at similar diversity levels. Conceptually, we believe that aggregating collective constraints in the MSA is the mechanism for property enhancement, but this remains to be shown.

6. Finally, I note that some of the figures contain a decent amount of small text. Please confirm with editorial staff that these will appear large enough to be easily readable.

We will work with the typesetter to ensure legibility.

Reviewer #3: In this report, Fram et al. describe the use of evolutionary models of sequence co-variation (EV couplings) to computationally design TEM1 beta lactamase variants with high sequence divergence from natural lactamases. In an initial step they generated a maximum entropy model, which includes site specific and pairwise constraints based on a multiple sequence alignment of the protein of interest. Using a sampling algorithm, which optimises fitness of the entire sequence, they could also define the sequence distance to the protein of interest as well as the nearest homologues. For testing their sequence design algorithms, they chose TEM1 beta lactamase, which is an ideal model system - as it is an intensely studied and well characterised model system. The authors generated several sequences for multiple sequence identity thresholds (98%, 95%, 90%, 80%, 70%, 50%). In addition, they also designed two additional mutants that were not constrained by sequence identity. Two designs for each sequence threshold, as well as the additional designs were experimentally tested and characterised. Out of the 14 designed sequences, 11 sequences conferred ampicillin resistance to *E. coli*, with 8 designs showing similar resistance as WT TEM-1 (70b, 50.a, 50.b being inactive). Additional growth assays showed that some of the designs are active against antibiotics, towards which WT-TEM1 is inactive.

The authors performed steady state kinetic analysis of the purified enzymes with nitrocefin as a substrate. Most designs showed either improvements in k_{cat} or K_M . Overall, multiple designs showed catalytic efficiencies that are higher than wild type TEM1. Using purified enzymes the authors also confirmed that the majority of their designs were capable of hydrolysing ampicillin with specific activities above WT-TEM1. Furthermore, the melting temperatures of their designs were increased by 10-20 °C compared to WT-TEM1. Structural analysis of 80.a, 80.b and 70.a by X-ray crystallography revealed that these proteins are in good agreement with WT-TEM1 and that structural deviations are in agreement with those of related lactamases with a similar level of sequence identity. Finally, the authors analysed the fitness contributions of individual mutations and showed that some of the introduced mutations are in fact on their own deleterious in WT TEM1 background, whereas in context of the design sequence they contribute to increased fitness.

Overall, this is a very interesting and very well executed piece of work. In silico sequence design and diversification of protein sequences is an important challenge in protein design. Various algorithms and design approaches with a similar goal have been implemented over time, including PROSS, Fireprot, ProteinMPNN or evolution based models (Ref.2), which are based on sequence alignments or structure prediction. The method outlined in this manuscript is a

valuable addition as the results presented in this work are impressive - up to 88 mutations were inserted into a natural protein, increasing protein stability, while maintaining enzyme activity. The manuscript is well written and the figures are well designed and informative.

I believe this work will be of interest to the broad readership of Nature Communications and I strongly recommend publication.

A few minor points need to be addressed:

1. In vitro ampicillin hydrolysis by opt.b was not determined as the authors claimed, this is due to opt.b being prone to precipitation. However Michaelis Menten parameters for nitrocefin could be measured for opt.b, suggesting that opt.b can be characterised in vitro. It would be thus desirable to determine the missing ampicillin hydrolysis value.

Apologies, and thanks for pointing out this confusing sentence. We were able to quantify ampicillin hydrolysis for opt.b, but the value was very low / close to the PBS-only condition and the limit of detection. The plotted data do include measured values for opt.b (they are just right on the zero line). We agree that it wasn't precipitation that likely caused the low activity measurement (i.e., the experiment was performed on soluble protein), and have removed the sentence from the text.

2. What were the protein expression levels (or yields)? These values are not mentioned in the manuscript. In addition, it is mentioned that 50.a and 50.b could not be purified - does this mean they could not be expressed?

We purified the constructs multiple times throughout the project and the yields were variable depending on culture conditions. We look at gels of both the soluble and insoluble fraction, and so we know that the 50.a and 50.b are expressed, but are not soluble.

3. Melt curves and Michaelis Menten plots are missing in the SI.

Thank you for pointing this out. Those data were available in the scripts released on GitHub, but agree they should be in the text and have added these plots to the supplement (Supplemental Figures S8, S9, S10).

4. Regarding the in vivo resistance assays, would it be possible to perform measurements at higher antibiotic concentrations, as WT-TEM1 and several of the designs show resistance at the highest chosen ampicillin concentration? The designs might compare even more favourable to the WT-TEM1.

When these experiments were originally performed, we did attempt to hone in on a more exact MIC, but experiments at higher concentrations tended to have reproducibility issues, especially between assay types (for unknown reasons).

5. It is unclear what the authors mean by "saturated concentration of ampicillin". 800 μ M does not seem saturated. Also, in the method section "Ampicillin" there seems to be a mistake, as parts of the nitrocefin section are repeated.

We agree and have removed the word saturated. During assay setup this was the highest concentration we used, and was also one in which the measurements were consistent. Also, thank you for pointing out the mistake in the methods section for the ampicillin data analysis, we have updated it with the correct text.

6. Figure 3C: The units on the y-axis are missing.

Thank you for catching this. We have updated the figure.

7. In the last part of the paper, the authors computationally analyse the contribution to fitness of the designed mutations and compare them to the results of a deep scanning experiment. Additional experimental data (characterisation of a few point mutants) could further confirm predictions/conclusions and validate their approach, making this an even stronger manuscript.

Thank you for the suggestion. Although we agree it could strengthen the manuscript, at this stage we feel it is beyond the scope of the manuscript.

8. I encourage the authors to cite a recent ProteinMPNN preprint from the Baker lab in their manuscript: <https://doi.org/10.1101/2023.10.03.560713>

We have added the citation in the manuscript.

9. The criteria for choosing the designs for characterisation should be mentioned in the main text - not the methods (Sequence subsampling). In this regard, was the 90% criteria chosen, to ensure that the catalytic residues are maintained?

This question led us to discover an error in the text. During the initial planning stages we had run through a list of criteria for how to subsample for testing (as was in the initial submission), however upon revisiting our notes we found that we decided to randomly pick two designs from each of the six. Apologies for the confusion and error.

10. With regard to point 9, the question also raises, how efficient is the use of EV-couplings in maintaining the catalytic residues? It was surprising that the neg. ctrl. S70A did not have a devastating effect on predicted fitness. What about other active site residues, were these conserved in the designs? It would be worthwhile to also indicate active site positions in figure 2C or 2B (e.g. sequence logo). Overall, how well does the presented method conserve active site residues and therefore activity - or is calculated fitness primarily based on structure/stability.

Although it appears that S70A does not have a strong influence on predicted fitness when shown together sequences that contain many amino acid differences, this is likely an artifact of

the predicted fitness calculation that arises at different mutational depths (see Shaw et al., bioRxiv, <https://doi.org/10.1101/2023.09.28.560044>). When only looking at a distribution of all single point mutants in WT TEM-1, it is more clear that S70A, E166A have a strong negative effect on fitness:

Mutations to catalytic residues S70A and E166A are predicted to have a strong negative effect on fitness while stabilizing mutation M182T is predicted to be one of the few mutations that have a positive fitness effect.

Conceptually, the model conforms to the MSA and will recapitulate conservation and interactions to the extent that they exist in the alignment. So if the homologs are not functionally similar enough, one can expect to generate mutations in the active site. Conversely, if the active site is conserved, the design residues should be conserved as well.

Closer examination of all highly invariant positions (those that have the same amino acid in >90% of MSA sequences), reveals that 32 of the 36 distance constrained designs have no mutations in these 10 positions. The four sequences that did mutate a highly-conserved position all had the same K234R mutation, which may or may not be functional (only 50.a was a tested design and it was non-functional, but it also had an additional 130 mutations). All of the distance constrained designs also maintain S70 and E166. This is interesting because although S70 is in ~94% of the MSA sequences used for model inference, E166 is only in ~70% of the same sequences - suggesting that our MSA might contain a number of sequences that are not beta lactamases (or have alternative mechanisms of catalysis, etc). It is tempting to speculate that this may indicate the design process and fitness-effect prediction is relatively robust to “contaminating” sequences, and that generated designs will generally conform to the function of the “majority” of the MSA. However, more research would be needed to say anything conclusive.

11. Every sequence contained the M182T mutation, which contributes to stability in WT-TEM1. Was this coincidence? Did all of the six designs of each threshold have this mutation? Or was there potential bias in the MSA. What is the fitness prediction of this mutation (SI Fig. 1 suggest that there only very few single mutants with increased predicted fitness)? Does EV-coupling primarily report on structure rather than activity?

Very interesting observation. We do not believe this to be a coincidence as M182T is one of the few mutations that is predicted to have a positive effect on fitness in WT TEM-1 (see figure in previous question #10 above). All but one of the 36 distance-constrained designs (including those not tested) had M182T. The design that did not have M182T was one of the 50% designs with 131 mutations total. We can only speculate as to whether EVcouplings primarily reports on structure rather than activity, but have seen that the model can be used to predict both. The model is derived from the homology alignment, and homology-based alignments are largely driven by stable natural sequences and have similar structure, and quite often have similar functions. Of course not all alignments meet that statement. In the MSA used in this work, the majority likely catalyze beta lactam hydrolysis, but a sizable fraction may not (see question #10 above). It is however plausible that these sequences have the same backbone structure as the beta lactamases. These "contaminants" may not always be a disadvantage, for example they may enable more diverse stabilizing mutations as long as the catalytic residues are not affected. However, it is likely highly context dependent and additional research would be required to determine how the structure/function makeup of the MSA affects design diversity and catalytic activity.

12. k_{cat} , K_M and T_m are incorrectly formatted throughout the manuscript.

Thank you for pointing this out. We have updated the manuscript with the correct formatting.

13. In the analysis of the X-ray structures towards rationalising T_m increases, the authors tallied the number of hydrogen bonds and atom pair contacts but could not find significant differences. Did the authors analyse the B-factors of the structures?

We did not in the original submission, but have now taken a closer look at B-factors as well as residue surface accessibility. See Reviewer #2 Question #2 for more details.

14. Introduction: Directed evolution is not a "common approach to protein design" - rather the contrary. It might also be worthwhile to include that directed evolution is costly and time consuming.

We have updated the text.

15. Figure 2: A: there are gaps on the blue bars at around position 58 and 239 - is this intentional? Furthermore, the description of the legend is unclear. Are the blue dots PDB contacts that coincide with model couplings? B: How are core residues defined? C: There are

gaps in the sequence logo and the conservation plot - was this intentional? What is the meaning of lower-case letters in the WT sequence?

(A) Those gaps are intentional as they indicate positions that are not aligned in the multiple sequence alignment. Unfortunately, due to scaling and color choice it is hard to parse in the original figure. We have updated the figure to contain a new color and continuous bar to avoid the scaling issues, and have also modified the legend and caption to make this clear. (B) Core residues were defined by their relative surface accessibility in the WT TEM-1 structure 1XPB. We have updated the figure caption to include this detail. (C) The gaps in the sequence logo represent unaligned positions in the multiple sequence alignment (as does the bar in the contact map). These unaligned positions are also depicted in lowercase in the MSA. We have updated the figure caption to make this clear.

16. The kinetic parameters for nitrocefin are likely below those of ampicillin for WT-TEM1, as nitrocefin is not the actual substrate. Is there any data in the literature that compares/correlates the ampicillin to nitrocefin hydrolysis in WT-TEM1?

We agree that it is unlikely that Nitrocefin is a native substrate for beta lactamases in general. It is however a useful substrate for determining the ability of an enzyme to catalyze beta lactam hydrolysis. It is easy, consistent, well documented in the literature, and generates reliable kinetic measurements. However, the magnitude and relevance of the resulting kinetics to native substrates is debatable, especially given that some have strong substrate preferences rather than just "good" or "bad" activity across the board. We searched for an analysis that compares ampicillin to nitrocefin kinetics, but we were unable to find any notable literature.

17. SI Figure 8: It is unclear to what "functional design" refers to in the legend?

Thank you for catching this omission. We have updated the text to clarify which designs are included in the definition (i.e., 70.a, 80.a, 80.b, 90.a, 90.b, 95.a, 95.b, 98.a, 98.b, opt.a, opt.b).

18. It would be desirable, if the authors could discuss in the manuscript, why consensus design and EV-couplings increases protein stability.

We have added some commentary into the discussion.

19. Additional images of structural alignments in the SI would be desirable, to highlight differences to WT-TEM1 (as figure 6C is hard to read), in particular of the active site to highlight the mutations that enable activity towards aztreonam.

We have added a new figure highlighting the active site and nearby mutations (which are discussed in the main text) for each of the design structures (Supplemental Figure 21).

20. The summaries after each paragraph are unnecessary and should be moved to the conclusion section.

We will work with the editor to make sure we conform with the journal style, and move to the discussion as needed.

21. Sequences of the construct (e.g. DNA sequences) are not included in the SI.

We agree, and will make sure they are included with the online supplemental information as well as in the GitHub repository.

Reviewer #4: Fram et al. present a highly interesting study wherein evolutionary information was used to design variants of TEM1 beta lactamase with enhanced stability and broadened substrate specificity. They computationally constructed and experimentally characterized fourteen variants of TEM-1 B-lactamase with varying sequence identity to the native TEM-1, and showed that the vast majority of these variants maintained activity while increasing stability. The work is an important addition to the somewhat limited literature that experimentally tests the capacity of evolution-based design algorithms. The experiments are well done, and it is nice to see the accompanying structures. Critically their results show both where the method succeeds and breaks down. The results suggest high potential for a “jump and walk” laboratory evolution strategy, wherein protein designs are used as new starting points for the evolution of enhanced or novel function. The manuscript is nicely written, the data analysis is rigorously presented, and the accompanying github repository appears reasonably well-organized. We anticipate that this work will be of broad interest to the protein engineering and design community, some enzyme biochemists, and those specifically working on the TEM-1 model system. While we are enthusiastic about the work as a whole, we have two major concerns and a few minor concerns.

Major concerns:

1. The authors state that “surprisingly, all functional designs had large increases in thermostability”, which seems to overlook prior work suggesting that evolutionary models (particularly Potts-like models, including DCA and EVcouplings) show some correlation between favorable score and increased thermodynamic stability. For example, see: Morcos et al (2014) “Coevolutionary information, protein folding landscapes, and the thermodynamics of natural selection” PNAS 111:12408. And even more critically: Figliuzzi et al (2015) “Coevolutionary landscape inference and the context dependence of mutations in beta-lactamase TEM-1” Mol Biol Evol 33:268. Figliuzzi et al. constructs a highly similar Potts-style model of the TEM-1 beta-lactamase family, and in figure 4 shows some correlation between model score and thermal stability. These prior results support and indeed almost predict the authors’ results that evolutionary models resulted in more thermal stable designed proteins. It would improve the manuscript to more thoroughly discuss the connection between this earlier work and the authors’ own findings. It would likewise be interesting to hear any ideas from the authors about why EVcouplings designs are more stable. The authors found that the increased thermal stability is not explained by increased hydrogen bonding, and they found that all designs

contained known stability-enhancing mutants. Additional questions of interest could include: Does the TEM-1 alignment include a large fraction of sequences from thermophiles? Would removing thermophilic sequences from the alignment result in fewer stability-enhanced designs? Do the designs select mutations that improve the site or coupling parameters portion of the model score? Further commentary on how the EVcouplings model enhances stability may provide the authors an opportunity to contribute to the body of prior work in this area.

Thanks for pointing this out. Indeed it has been shown by multiple groups that delta EVH and other evolution-based model-predicted fitness correlates with thermostability, although the correlation is not perfect and has primarily been shown in relation to either natural sequences or sequences with a small number of mutations. We have removed the word surprising from the sentence, and now also point out in the main text that previous literature has observed this correlation. In addition, we have added more commentary to the discussion exploring the prior work and how it relates to our own findings.

Does the MSA contain many thermophiles? To answer this interesting question, we first need to identify the thermophilic sequences in the MSA used for model inference, which in itself is a non-trivial problem. To classify individual sequences we mapped each to their organism or origin and extracted taxonomy (taxon id). Although notably crude, only ~1.12% and ~0.94% of the organism names contain “therm” or “thermo”, respectively. Examination of overlap of the taxon ids with the ThermoBase database (ver 1.0, 2022) found an overlap of ~0.79%. Using the SCMTPP thermophile predictor (<https://www.nature.com/articles/s41598-021-03293-w>) approximately 1.7% of the sequences were classified as thermophilic. These results suggest that the model inference was not heavily influenced by thermophilic sequences, at least likely not enough to substantially influence the predicted fitness scores. Lastly, we manually examined the top five homologs to each of the designs, and none were from thermophiles, suggesting that the model did not push the designs towards any specific thermophiles in the MSA used for inference. We believe that the more likely explanation for increased stability has to do with aggregating stabilizing mutations from the diverse sequences (see Discussion).

Do the designs select mutations that improve the site or coupling parameters portion of the model score? When looking at the independent (site-only) model, the predicted fitness of active characterized designs is much less accurate than the combined sitewise and couplings model (e.g., the non-functional designs 50.a, 50.b, and rw-consensus, all had some of the highest predicted fitness), suggesting that the couplings are essential for our design methodology. However, numerically the sitewise parameters do heavily influence these scores, and both the sitewise and couplings play an important role in the overall predicted fitness.

2. One aspect of evolution-informed computational protein design is the choice of algorithm for inferring site and coupling parameters given a sequence alignment. EVcouplings uses regularized maximum pseudolikelihood, however other strategies exist (mean field, ACE, Boltzmann Machine). These strategies vary in the fidelity with which they recapitulate pairwise (and higher order) statistics of the natural alignment (e.g. Fig 2 of Cocco et al. (2018) “Inverse

statistical physics of protein sequences: a key issues review” Reports on Progress in Physics). How well does the inferred EVcouplings model recapitulate the pairwise alignment statistics of the natural alignment (i.e. if you compare between an alignment of designed and natural sequences)? It would be nice to have a plot of this alongside the other model validation strategies (showing recovery of contacts and prediction of mutant effects). Additionally, could the authors spend some time to discuss what might be improved to push designs beyond the apparent 70% identity threshold? Is this limitation coming from the inference strategy, sampling size limitations, a need for higher order statistics, or perhaps something else entirely?

Thank you for the thought-provoking comment. Unfortunately, we do not have enough designed sequences that have been characterized to infer a new model and compare with the model inferred from natural homologs, which would require thousands of characterized sequence designs. Given the sampling methodology we applied (maximizing predicted fitness and model score), we are doubtful that the designs would recapitulate the natural alignment statistics. This is not due to the model itself, but rather because our goal was not to match the distribution of natural predicted fitness scores. However, previous work has probed this question and offers some insight into whether these EVcouplings-style models are able to recapitulate the natural alignment statistics: the Ranganathan lab applied bmDCA to chorismate mutase from natural sequences, derived and tested variant sequences using the predicted fitness from the model, and demonstrated that bmDCA could effectively re-generate equivalent sequence distributions as those from the natural alignment (Russ et al., 2020).

What is the limit of the model and how do we push beyond 70%? We actually don't believe there is a 70% threshold, but it is likely that we just tested too few sequences to see activity with larger mutational burdens. However even if there is no hard threshold, the fitness predictions likely do become worse with increasing mutations as (1) sequences become even more different from the MSA on which the model was parameterized due to distance constraint requirements in the objective function and (2) there are fewer unmutated positions allowed, and the model must begin to sample the less mutation-tolerant positions. The limitations you mention (sampling size, etc) may all play a role. To reach maximum diversity from the starting WT sequence, a viable strategy might be to relax the distance constraint to the MSA. Although speculative, if the goal is to generate sequences maximally different from known sequences, then it likely requires sampling a larger or more functionally-well defined MSA.

Minor concerns:

1. It would be useful to indicate the number of designed sequences that were experimentally characterized in the abstract

We have updated the abstract.

2. Again in the abstract: this is probably more a matter of taste, but I would not typically consider a sequence with 70% identity to the WT as “highly divergent”. Maybe “highly mutated” or “with many mutations”?

Thanks for the suggestion. We agree and have updated the text.

3 .On page 4, a sentence or two briefly explaining greedy sampling and parallel tempering would help a broader audience with less expertise in optimization.

We agree and have added additional text.

4 .On page 10, 9 lines from bottom, “an 7.5C increase” should be changed to “a 7.5C increase”

Thank you for catching this typo. We have updated the text.

5 .Please provide an equation for the Gibbs sampling cost function in the methods section

Thank you for pointing out this oversight. We have added the equation and more details of the design generation to the methods section.

6. Supplemental figure 3b legend states that “Residues mutated in the designs were less likely to be surface accessible than non-mutated residues”. This seems to contradict the figure and what is written in the main text (“the algorithm preferred mutating positions that were closer to the surface”)

Thank you for catching this typo. We have updated the text.

7. It would be helpful to indicate the active site in Figure S4.

We have updated the figure to highlight the catalytic residues (now Supplemental Figure S6).

Reviewer Response Addendum Figure 1 - Solvent Accessible Surface Area (SASA) analysis. Relationship of absolute solvent accessible surface areas (SASA) and relative SASA of individual residues in the crystal structures of WT TEM-1 (PDB: 1XPB, 4GKU, and 1S0W) and the designs 70.a, 80.a, and 80.b. Absolute SASA values (in Å²) were calculated using the DSSP program. Relative SASA (as percentage) were calculated by normalizing the absolute SASA by empirical maximum SASA values published in Tien et al., PLoS One, 2013 (PMC3836772). Relative SASA denoted on the x-axis and y-axis labels (otherwise SASA is absolute). Residues in the active site pocket are highlighted as open circles. Active site pocket is defined as residues within 4 Å from the catalytic residues S70 and E166. Although there were differences for individual residues, none of the differences was consistent (i.e., had the same difference between all WT structures and the design).

Reviewer Response Addendum Figure 2 - B-factor analysis. Relationship of B-factors of individual residues (C α -atom only and average of sidechain atoms) in the crystal structures of WT TEM-1 (PDB: 1XPB, 4GKU, and 1S0W) and the designs 70.a, 80.a, and 80.b. Residues in the active site pocket are highlighted as open circles. Active site pocket is defined as residues within 4 Å from the catalytic residues S70 and E166. Although there were individual residue differences, none of these differences was consistent between all WT structures and the design structure.

Reviewers' Comments:

Reviewer #1:

Remarks to the Author:

The authors addressed my concerns.

Reviewer #2:

Remarks to the Author:

I have reviewed the authors' responses to the reviewer comments, and I am satisfied that all significant concerns have been addressed. I believe the manuscript should be accepted for publication in this revised form.

In particular, I appreciate the authors attention to my technical comments about crystallography and structure comparisons. I will note that in the presence of resolution differences, one can compare the Z-scores of B-factors, rather than the B-factors themselves, which can often overcome the effect of resolution differences. Nevertheless, the authors make a good point regarding the comparisons of their structures to other preexisting ones, and I don't think it is worth performing any additional calculations at this point as they are unlikely to show anything useful.

Reviewer #3:

Remarks to the Author:

The authors have addressed the raised points. I would also like to apologize, as I realized that the data on ampicillin hydrolysis of opt.b was in the first version of the manuscript. Unfortunately, the quality of my print-out made it look like there was no data point.

However, a few minor points remain:

- The sentence on precipitation is still in the revised manuscript ("We found purified opt.b unstable and prone to precipitation, which may explain the discrepancy between bacterial and biochemical assays.").

- The kinetic plots were added to the SI, however the Michaelis-Menten fit (overlying the data points) is missing. A table with all the values would be a useful addition to the SI as well. Regarding analysis, many of the mutants seem to show substrate inhibition (e.g. 95a, 80a, 70a etc.). It would be desirable to use Michaelis-Menten analysis with substrate inhibition. Omitting the 800 μ M value for fitting is not suitable for some variants as the highest substrate concentration should be at least 3-fold above the K_m (e.g. 400 μ M as the highest substrate concentration only allows reliable MM analysis of variants with a K_m below 135 μ M). Following up from this, how do the kinetic parameters for nitrocefin of the WT-TEM1 compare to literature values?

- The melt curves are missing in the SI. On the GitHub repository, the 'raw' data is not directly accessible (only the bar chart). I would recommend including the plots (x: temperature, y: fluorescence) of the differential scanning fluorimetry in the SI to show the melting behaviour (e.g. is it a single transition?).

- Regarding the units of the y-axis in 3C: kcat is min⁻¹ not μ M/min, kcat/ K_m is M⁻¹min⁻¹ and not min⁻¹.

- In my last review, I suggested not to call directed evolution 'a common approach to protein design'. In their revised manuscript, the authors still refer to directed evolution as protein design ("One approach to protein design, directed evolution, makes use of iterative rounds [...]"). I would encourage them to replace 'design' by 'engineering'. Design infers a rational/computational approach, while directed evolution is an 'agnostic' approach

that does not require any rationale or understanding.

Besides these minor points, publication is recommended.

Reviewer #4:

Remarks to the Author:

The authors' revisions have satisfied all of my concerns. Congratulations on a nice manuscript!

Reviewer #5:

Remarks to the Author:

Simultaneous Enhancement of Multiple Functional Properties Using Evolution-informed Protein Design

Point-by-point responses to reviewers' comments:

We again thank the reviewers for the constructive set of comments that have improved the manuscript.

Reviewer #1 (Remarks to the Author):

The authors addressed my concerns.

Reviewer #1 (Remarks on code availability):

The code now works.

Reviewer #2 (Remarks to the Author):

I have reviewed the authors' responses to the reviewer comments, and I am satisfied that all significant concerns have been addressed. I believe the manuscript should be accepted for publication in this revised form.

In particular, I appreciate the authors attention to my technical comments about crystallography and structure comparisons. I will note that in the presence of resolution differences, one can compare the Z-scores of B-factors, rather than the B-factors themselves, which can often overcome the effect of resolution differences. Nevertheless, the authors make a good point regarding the comparisons of their structures to other preexisting ones, and I don't think it is worth performing any additional calculations at this point as they are unlikely to show anything useful.

Reviewer #3 (Remarks to the Author):

The authors have addressed the raised points. I would also like to apologize, as I realized that the data on ampicillin hydrolysis of opt.b was in the first version of the manuscript. Unfortunately, the quality of my print-out made it look like there was no data point.

However, a few minor points remain:

- The sentence on precipitation is still in the revised manuscript ("We found purified opt.b unstable and prone to precipitation, which may explain the discrepancy between bacterial and biochemical assays.").

Apologies, that change was likely lost when aggregating comments. We have removed the sentence.

- The kinetic plots were added to the SI, however the Michaelis-Menten fit (overlying the data points) is missing. A table with all the values would be a useful addition to the SI as well. Regarding analysis, many of the mutants seem to show substrate inhibition (e.g. 95a, 80a, 70a etc.). It would be desirable to use Michaelis–Menten analysis with substrate inhibition. Omitting the 800µM value for fitting is not suitable for some variants as the highest substrate concentration should be at least 3-fold above the Km (e.g. 400µM as the highest substrate concentration only allows reliable MM analysis of variants with a Km below 135µM). Following up from this, how do the kinetic parameters for nitrocefin of the WT-TEM1 compare to literature values?

We have made multiple changes from these very helpful tips:

- (1) Supplemental Figure 10 now contains the overlay of the parameter fit to the measured data (initial reaction rate as a function of substrate concentration), and we have added a Supplemental Table S1 with the values.
- (2) The data were re-fit to the Michaelis-Menten equation with the 800 µM substrate concentration included when the Km (at 400 µM) was greater than ~133 µM (1/3 the 400 µM concentration). These designs are WT TEM-1, 98.b, 90.b, opt.b. The two exceptions we made to this criteria were opt.a and 70.a, which both clearly had substrate inhibition at 800 µM (for opt.a and 70.a enzymes the Km was slightly over the 1/3 criteria at ~140 µM and 163 µM, respectively). Also note that even when including the maximum measured (800 µM) substrate, the Km of the opt.b construct (~343 µM) is over the 3x criteria ($3 \times 343 \approx 1029$). Although we observe small magnitude changes in the absolute Km and kcat due to this analysis change, overall the trends relative to WT TEM-1 changed little:

Original analysis:

Updated analysis (including 800 µM substrate for WT TEM-1, 98.b, 90.b, 70.a, opt.b):

- (3) Inhibition: We attempted to fit the Michaelis-Menten equation with substrate inhibition, but observed large uncertainties on the fitting parameters for nearly all the designs. This may be due to a lack of data points in the high substrate concentration region. The analyses are included in the GitHub code, but we have decided to leave them out of the main manuscript.

(4) There are not as many references to WT TEM-1 hydrolysis of nitrocefin as we expected, but we did find one report that details Michaelis-Menten kinetics on WT TEM-1 (<https://www.ncbi.nlm.nih.gov/pmc/articles/PMC4400348/>). We observe approximately an order of magnitude difference between our results:

	K_m (μM)	K_{cat} (sec^{-1})	K_m/K_{cat} ($\mu\text{M}^{-1}\text{x sec}^{-1}$)
Stojanoski et al.	30 ± 10	714 ± 80	24
This study	136 ± 18	228 ± 10	1.7
abs(Difference)	~4.5X	~3.1X	~14.1X

We interpret the differences as likely due to different enzyme preparations, reagents etc.

- The melt curves are missing in the SI. On the GitHub repository, the 'raw' data is not directly accessible (only the bar chart). I would recommend including the plots (x: temperature, y: fluorescence) of the differential scanning fluorimetry in the SI to show the melting behaviour (e.g. is it a single transition?).

Apologies, we misunderstood this request from the first revision. We have added the additional figure to the supplement and the raw fluorescence-temperature to the datafile.

- Regarding the units of the y-axis in 3C: kcat is min^{-1} not $\mu\text{M}/\text{min}$, kcat/K_m is $\text{M}^{-1}\text{min}^{-1}$ and not min^{-1} .

Thank you for catching this error. We have updated the figure.

- In my last review, I suggested not to call directed evolution 'a common approach to protein design'. In their revised manuscript, the authors still refer to directed evolution as protein design ("One approach to protein design, directed evolution, makes use of iterative rounds [...]"). I would encourage them to replace 'design' by 'engineering'. Design infers a rational/computational approach, while directed evolution is an 'agnostic' approach that does not require any rationale or understanding.

Although we are not in full agreement with this comment, we have updated the text to "engineering" as this could be controversial.

Besides these minor points, publication is recommended.

Reviewer #4 (Remarks to the Author):

The authors' revisions have satisfied all of my concerns. Congratulations on a nice manuscript!

Reviewer #4 (Remarks on code availability):

The code is reasonably well documented and organized.

Reviewer #5 (Remarks to the Author):

Reviewers' Comments:

Reviewer #3:

In their revised version, the authors included the proposed text changes and corrections to axis labelling in the main text. Furthermore, in the SI they included the Michaelis-Menten fits in the kinetic plots and the fluorescence raw data of the melt curves.

Overall, their kinetic data supports their claims that most designs show activities that are similar to WT-TEM1. However, considering the neglect of substrate inhibition during Michaelis-Menten fitting (which would result in lowered K_m values, as well as lowered k_{cat} values), as well as the deviation of kinetic parameters of WT-TEM1 from literature values, I would be hesitant to refer to the designs as better than WT-TEM1. (Furthermore, there is no mention of substrate inhibition in the main text.)

"All of the designs that enabled resistance to ampicillin in the biological assays had a similar or elevated catalytic efficiency (k_{cat}/K_M) to WT TEM-1."

"Interestingly, many of the changes in catalytic efficiency are driven by changes to K_M , with the majority of the designs showing a decreased K_M relative to WT TEM-1, likely reflecting increased binding affinity to nitrocefin."

 All variants with decreased K_m also show substrate inhibition, suggesting that the lowered K_m values are an artifact of fitting without substrate inhibition.

Simultaneous Enhancement of Multiple Functional Properties Using Evolution-informed Protein Design

Point-by-point responses to reviewers' comments:

We again thank the reviewers for the constructive set of comments that have improved the manuscript.

Reviewer #3

In their revised version, the authors included the proposed text changes and corrections to axis labelling in the main text. Furthermore, in the SI they included the Michaelis-Menten fits in the kinetic plots and the fluorescence raw data of the melt curves.

Overall, their kinetic data supports their claims that most designs show activities that are similar to WT-TEM1. However, considering the neglect of substrate inhibition during Michaelis Menten fitting (which would result in lowered K_m values, as well as lowered k_{cat} values), as well as the deviation of kinetic parameters of WT-TEM1 from literature values, I would be hesitant to refer to the designs as better than WT-TEM1. (Furthermore, there is no mention of substrate inhibition in the main text.)

"All of the designs that enabled resistance to ampicillin in the biological assays had a similar or elevated catalytic efficiency (k_{cat}/K_M) to WT TEM-1."

"Interestingly, many of the changes in catalytic efficiency are driven by changes to K_M , with the majority of the designs showing a decreased K_M relative to WT TEM-1, likely reflecting increased binding affinity to nitrocefin."

 All variants with decreased K_m also show substrate inhibition, suggesting that the lowered K_m values are an artifact of fitting without substrate inhibition.

We have removed the language relating to the decrease in K_M and the text indicating better performance compared to WT TEM-1. In addition, we have now noted the substrate inhibition in the main text.